# AUTOMATED DATA AUGMENTATIONS FOR GRAPH CLASSIFICATION

**Youzhi Luo**[1][*]**, Michael McThrow**[2]**, Wing Au**[2]**, Tao Komikado**[3]**, Kanji Uchino**[2]**,
Koji Maruhashi**[3]**, Shuiwang Ji**[1]
[1]Texas A&M University, TX, USA
[2]Fujitsu Research of America, INC., CA, USA
[3]Fujitsu Research, Fujitsu Limited, Kanagawa, Japan
`{yzluo,sji}@tamu.edu`
`{mmcthrow,WAu,komikado.tao,kanji,maruhashi.koji}@fujitsu.com`

## ABSTRACT

Data augmentations are effective in improving the invariance of learning machines. We argue that the core challenge of data augmentations lies in designing data transformations that preserve labels. This is relatively straightforward for images, but much more challenging for graphs. In this work, we propose GraphAug, a novel automated data augmentation method aiming at computing label-invariant augmentations for graph classification. Instead of using uniform transformations as in existing studies, GraphAug uses an automated augmentation model to avoid compromising critical label-related information of the graph, thereby producing label-invariant augmentations at most times. To ensure label-invariance, we develop a training method based on reinforcement learning to maximize an estimated label-invariance probability. Experiments show that GraphAug outperforms previous graph augmentation methods on various graph classification tasks.

## 1 INTRODUCTION

Many real-world objects, such as molecules and social networks, can be naturally represented as graphs. Developing effective classification models for these graph-structured data has been highly desirable but challenging. Recently, advances in deep learning have significantly accelerated the progress in this direction. Graph neural networks (GNNs) (Gilmer et al., 2017), a class of deep neural network models specifically designed for graphs, have been widely applied to many graph representation learning and classification tasks, such as molecular property prediction (Wang et al., 2022b; Liu et al., 2022; Wang et al., 2022a; 2023; Yan et al., 2022).

However, just like deep models on images, GNN models can easily overfit and fail to achieve satisfactory performance on small datasets. To address this issue, data augmentations can be used to generate more data samples. An important property of desirable data augmentations is label-invariance, which requires that label-related information should not be compromised during the augmentation process. This is relatively easy and straightforward to achieve for images (Taylor & Nitschke, 2018), since commonly used image augmentations, such as flipping and rotation, can preserve almost all information of original images. However, ensuring label-invariance is much harder for graphs because even minor modification of a graph may change its semantics and thus labels. Currently, most commonly used graph augmentations (You et al., 2020) are based on random modification of nodes and edges in the graph, but they do not explicitly consider the importance of label-invariance.

In this work, we propose GraphAug, a novel graph augmentation method that can produce label-invariant augmentations with an automated learning model. GraphAug uses a learnable model to automate augmentation category selection and graph transformations. It optimizes the model to maximize an estimated label-invariance probability through reinforcement learning. Experimental results show that GraphAug outperforms prior graph augmentation methods on multiple graph classification tasks. The codes of GraphAug are available in DIG (Liu et al., 2021) library.

---

[*]Work was done while the author was at Fujitsu Research of America, INC.

## 2 BACKGROUND AND RELATED WORK

### 2.1 GRAPH CLASSIFICATION WITH NEURAL NETWORKS

In this work, we study the problem of graph classification. Let $G = (V, E, X)$ be an undirected graph, where $V$ is the set of nodes and $E$ is the set of edges. The node feature matrix of the graph $G$ is $X \in \mathbb{R}^{|V| \times d}$ where the $i$-th row of $X$ denotes the $d$-dimensional feature vector for the $i$-th node in $G$. For a graph classification task with $k$ categories, the objective is to learn a classification model $f : G \to y \in \{1, ..., k\}$ that can predict the categorical label of $G$.

Recently, GNNs (Kipf & Welling, 2017; Veličković et al., 2018; Xu et al., 2019; Gilmer et al., 2017; Gao & Ji, 2019) have shown great success in various graph classification problems. Most GNNs use the message passing mechanism to learn graph node embeddings. Formally, the message passing for any node $v \in V$ at the $\ell$-th layer of a GNN model can be described as

$$h_v^\ell = \text{UPDATE}\left(h_v^{\ell-1}, \text{AGG}\left(\left\{m_{jv}^\ell : j \in \mathcal{N}(v)\right\}\right)\right), \tag{1}$$

where $\mathcal{N}(v)$ denotes the set of all nodes connected to the node $v$ in the graph $G$, $h_v^\ell$ is the embedding outputted from the $\ell$-th layer for $v$, $m_{jv}^\ell$ is the message propagated from the node $j$ to the node $v$ at the $\ell$-th layer and is usually a function of $h_v^{\ell-1}$ and $h_j^{\ell-1}$. The aggregation function $\text{AGG}(\cdot)$ maps the messages from all neighboring nodes to a single vector, and the function $\text{UPDATE}(\cdot)$ updates $h_v^{\ell-1}$ to $h_v^\ell$ using this aggregated message vector. Assuming that the GNN model has $L$ layers, the graph representation $h_G$ is computed by a global pooling function READOUT over all node embeddings as

$$h_G = \text{READOUT}\left(\left\{h_v^L : v \in V\right\}\right). \tag{2}$$

Afterwards, $h_G$ is fed into a multi-layer perceptron (MLP) model to compute the probability that $G$ belongs to each of the categories $\{1, ..., k\}$.

Despite the success of GNNs, a major challenge in many graph classification problems is data scarcity. For example, GNNs have been extensively used to predict molecular properties from graph structures of molecules. However, the manual labeling of molecules usually requires expensive wet lab experiments, so the amount of labeled molecule data is usually not large enough for expressive GNNs to achieve satisfactory prediction accuracy. In this work, we address this data scarcity challenge with data augmentations. We focus on designing advanced graph augmentation strategies to generate more data samples by performing transformations on data samples in the dataset.

### 2.2 DATA AUGMENTATIONS

Data augmentations have been demonstrated to be effective in improving the performance for image and text classification. For images, various image transformation or distortion techniques have been proposed to generate artificial image samples, such as flipping, cropping, color shifting (Krizhevsky et al., 2012), scaling, rotation, and elastic distortion (Sato et al., 2015; Simard et al., 2003). And for texts, useful augmentation techniques include synonym replacement, positional swaps (Ratner et al., 2017a), and back translation (Sennrich et al., 2016). These data augmentation techniques have been widely used to reduce overfitting and improve robustness in training deep neural network models.

In addition to hand-crafted augmentations, automating the selection of augmentations with learnable neural network model has been a recent emerging research area. Ratner et al. (2017b) selects and composes multiple image data augmentations using an LSTM (Hochreiter & Schmidhuber, 1997) model, and proposes to make the model avoid producing out-of-distribution samples through adversarial training. Cubuk et al. (2019) proposes AutoAugment, which adopts reinforcement learning based method to search optimal augmentations maximizing the classification accuracy. To speed up training and reduce computational cost, a lot of methods have been proposed to improve AutoAugment through either faster searching mechanism (Ho et al., 2019; Lim et al., 2019), or advanced optimization methods (Hataya et al., 2020; Li et al., 2020; Zhang et al., 2020).

### 2.3 DATA AUGMENTATIONS FOR GRAPHS

While image augmentations have been extensively studied, doing augmentations for graphs is much more challenging. Images are Euclidean data formed by pixel values organized in matrices. Thus,

many well studied matrix transformations can naturally be used to design image augmentations, such as flipping, scaling, cropping or rotation. They are either strict information lossless transformation, or able to preserve significant information at most times, so label-invariance is relatively straight-forward to be satisfied. Differently, graphs are non-Euclidean data formed with nodes connected by edges in an irregular manner. Even minor structural modification of a graph can destroy important information in it. Hence, it is very hard to design generic label-invariant transformations for graphs.

Currently, designing data augmentations for graph classification (Zhao et al., 2022; Ding et al., 2022; Yu et al., 2022) is a challenging problem. Some studies (Wang et al., 2021; Han et al., 2022; Guo & Mao, 2021; Park et al., 2022) propose interpolation-based mixup methods for graph augmentations, and Kong et al. (2022) propose to augment node features through adversarial learning. Nonetheless, most commonly used graph augmentation methods (Hamilton et al., 2017; Wang et al., 2020; You et al., 2020; Zhou et al., 2020a; Rong et al., 2020; Zhu et al., 2021a) are based on the random modification of graph structures or features, such as randomly dropping nodes, perturbing edges, or masking node features. However, such random transformations are not necessarily label-invariant, because important label-related information may be randomly compromised (see Section 3.2 for detailed analysis and discussion). Hence, in practice, these augmentations do not always improve the performance of graph classification models.

## 3 THE PROPOSED GRAPHAUG METHOD

While existing graph augmentation methods do not consider the importance of label-invariance, we dive deep into this challenging problem and propose to solve it by automated data augmentations. Note that though automated data augmentations have been applied to graph contrastive learning (You et al., 2021; Yin et al., 2022; Suresh et al., 2021; Hassani & Khasahmadi, 2022; Xie et al., 2022) and node classification (Zhao et al., 2021; Sun et al., 2021), they have not been studied in supervised graph classification. In this work, we propose GraphAug, a novel automated data augmentation framework for graph classification. GraphAug automates augmentation category selection and graph transformations through a learnable augmentation model. To produce label-invariant augmentations, we optimize the model to maximize an estimated label-invariance probability with reinforcement learning. To our best knowledge, GraphAug is the first work successfully applying automated data augmentations to generate new graph data samples for supervised graph classification.

### 3.1 AUGMENTATION BY SEQUENTIAL TRANSFORMATIONS

Similar to the automated image augmentation method in Ratner et al. (2017b), we consider graph augmentations as a sequential transformation process. Given a graph $G_0$ sampled from the training dataset, we map it to the augmented graph $G_T$ with a sequence of transformation functions $a_1, a_2, ..., a_T$ generated by an automated data augmentation model $g$. Specifically, at the $t$-th step ($1 \leq t \leq T$), let the graph obtained from the last step be $G_{t-1}$, we first use the augmentation model to generate $a_t$ based on $G_{t-1}$, and map $G_{t-1}$ to $G_t$ with $a_t$. In summary, this sequential augmentation process can be described as

$$a_t = g(G_{t-1}), \quad G_t = a_t(G_{t-1}), \quad 1 \leq t \leq T. \tag{3}$$

In our method, $a_1, a_2, ..., a_T$ are all selected from three categories of graph transformations:

- **Node feature masking (MaskNF)**, which sets some values in node feature vectors to zero;
- **Node dropping (DropNode)**, which drops certain portion of nodes from the input graph;
- **Edge perturbation (PerturbEdge)**, which produces the new graph by removing existing edges from the input graph and adding new edges to the input graph.

### 3.2 LABEL-INVARIANT AUGMENTATIONS

Most automated image augmentation methods focus on automating augmentation category selection. For instance, Ratner et al. (2017b) automate image augmentations by generating a discrete sequence from an LSTM (Hochreiter & Schmidhuber, 1997) model, and each token in the sequence represents a certain category of image transformation, such as random flip and rotation. Following this setting,

our graph augmentation model $g$ also selects the augmentation category at each step. Specifically, $g$ will generate a discrete token $c_t$ representing the category of augmentation transformation $a_t$, denoting whether MaskNF, DropNode, or PerturbEdge will be used at the $t$-th step.

We have experimented to only automate augmentation category selection and use the graph transformations that are uniformly operated on each graph element, such as each node, edge, or node feature. For example, the uniform DropNode will randomly drop each node in the graph with the same probability. These transformations are commonly used in other studies (You et al., 2020; Zhu et al., 2021a; Rong et al., 2020), and we call them as **uniform transformations**. However, we find that this automated composition of multiple uniform transformations does not improve classification performance (see Section 4.3 for details). We argue that it is because uniform transformations have equal chances to randomly modify each graph element, thus may accidentally damage significant label-related information and change the label of the original data sample. For instance, in a molecular graph dataset, assuming that all molecular graphs containing a cycle are labeled as toxic because the cyclic structures are exactly the cause of toxicity. If we are using DropNode transformation, dropping any node belonging to the cycle will damage this cyclic structure, and map a toxic molecule to a non-toxic one. By default, data augmentations only involve modifying data samples while labels are not changed, so data augmentations that are not label-invariant may finally produce many noisy data samples and greatly harm the training of the classification model.

We use the TRIANGLES dataset (Knyazev et al., 2019) as an example to study the effect of label-invariance. The task in this dataset is classifying graphs by the number of triangles (the cycles formed by only three nodes) contained in the graph. As shown in Figure 3 of Appendix A, the uniform DropNode transformation is not label-invariant because it produces data samples with wrong labels through dropping nodes belonging to triangles, and the classification accuracy is low when the classification model is trained on these data samples. However, if we intentionally avoid dropping nodes in triangles, training the classification model with this label-invariant data augmentation improves the classification accuracy. The significant performance gap between these two augmentation strategies clearly demonstrates the importance of label-invariance for graph augmentations.

Based on the above analysis and experimental results, we can conclude that uniform transformations should be avoided in designing label-invariant graph augmentations. Instead, we generate transformations for each element in the graph by the augmentation model $g$ in our method. Next, we introduce the detailed augmentation process in Section 3.3 and the training procedure in Section 3.4.

## 3.3 AUGMENTATION PROCESS

Our augmentation model $g$ is composed of three parts. They are a GNN based encoder for extracting features from graphs, a GRU (Cho et al., 2014) model for generating augmentation categories, and four MLP models for computing probabilities. We use graph isomorphism network (GIN) (Xu et al., 2019) model as the encoder.

At the $t$-th augmentation step ($1 \leq t \leq T$), let the graph obtained from the last step be $G_{t-1} = (V_{t-1}, E_{t-1}, X_{t-1})$, we first add a virtual node $v_{virtual}$ into $V_{t-1}$ and add edges connecting the virtual node with all the nodes in $V_{t-1}$. In other words, a new graph $G'_{t-1} = (V'_{t-1}, E'_{t-1}, X'_{t-1})$ is created from $G_{t-1}$ such that $V'_{t-1} = V_{t-1} \cup \{v_{virtual}\}$, $E'_{t-1} = E_{t-1} \cup \{(v_{virtual}, v) : v \in V_{t-1}\}$, and $X'_{t-1} \in \mathbb{R}^{|V'_{t-1}| \times d}$ is the concatenation of $X_{t-1}$ and a trainable initial feature vector for the virtual node. We use the virtual node here to extract graph-level information because it can capture long range interactions in the graph more effectively than a pooling based readout layer (Gilmer et al., 2017). The GNN encoder performs multiple message passing operations on $G'_{t-1}$ to obtain $r$-dimensional embeddings $\{e^v_{t-1} \in \mathbb{R}^r : v \in V_{t-1}\}$ for nodes in $V_{t-1}$ and the virtual node embedding $e^{virtual}_{t-1} \in \mathbb{R}^r$. Afterwards, the probabilities of selecting each augmentation category is computed from $e^{virtual}_{t-1}$ as $q_t = \text{GRU}(q_{t-1}, e^{virtual}_{t-1})$, $p^C_t = \text{MLP}^C(q_t)$, where $q_t$ is the hidden state vector of the GRU model at the $t$-th step, and the MLP model $\text{MLP}^C$ outputs the probability vector $p^C_t \in \mathbb{R}^3$ denoting the probabilities of selecting MaskNF, DropNode, or PerturbEdge as the augmentation at the $t$-th step. The exact augmentation category $c_t$ for the $t$-th step is then randomly sampled from the categorical distribution with the probabilities in $p^C_t$. Finally, as described below, the computation of transformation probabilities for all graph elements and the process of producing the new graph $G_t$ from $G_{t-1}$ vary depending on $c_t$.

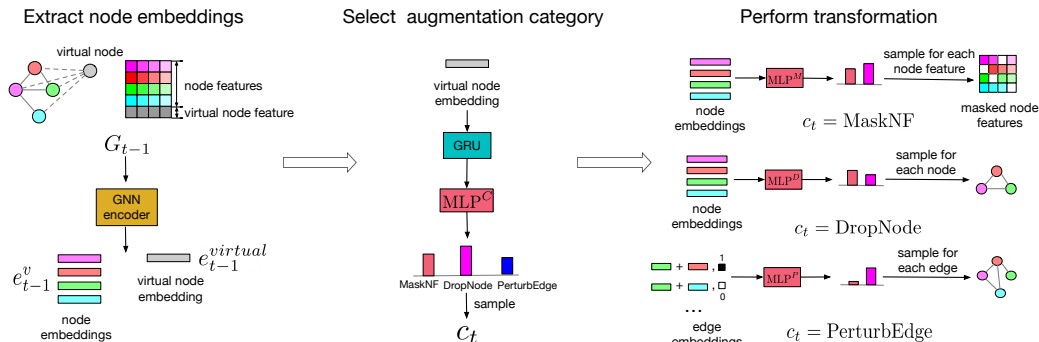

Figure 1: An illustration of the process of producing $G_t$ from $G_{t-1}$ with the augmentation model.

- If $c_t$ is MaskNF, then for any node $v \in V_{t-1}$, the probabilities $p_{t,v}^M \in \mathbb{R}^d$ of masking each node feature of $v$ is computed by the MLP model $\text{MLP}^M$ taking the node embedding $e_{t-1}^v$ as input. Afterwards, a binary vector $o_{t,v}^M \in \{0,1\}^d$ is randomly sampled from the Bernoulli distribution parameterized with $p_{t,v}^M$. If the $k$-th element of $o_{t,v}^M$ is one, i.e., $o_{t,v}^M[k] = 1$, the $k$-th node feature of $v$ is set to zero. Such MaskNF transformation is performed for every node feature in $X_{t-1}$.

- If $c_t$ is DropNode, then the probability $p_{t,v}^D$ of dropping any node $v \in V_{t-1}$ from $G_{t-1}$ is computed by the MLP model $\text{MLP}^D$ taking the node embedding $e_{t-1}^v$ as input. Afterwards, a binary value $o_{t,v}^D \in \{0,1\}$ is sampled from the Bernoulli distribution parameterized with $p_{t,v}^D$ and $v$ is dropped from $V_{t-1}$ if $o_{t,v}^D = 1$. Such DropNode transformation is performed for every node in $V_{t-1}$.

- If $c_t$ is PerturbEdge, the transformations involve dropping some existing edges from $E_{t-1}$ and adding some new edges into $E_{t-1}$. We consider the set $E_{t-1}$ as the droppable edge set, and we create an addable edge set $\overline{E}_{t-1}$, by randomly sampling at most $|E_{t-1}|$ addable edges from the set $\{(u,v) : u,v \in V_{t-1}, (u,v) \notin E_{t-1}\}$. For any $(u,v)$ in $E_{t-1}$, we compute the probability $p_{t,(u,v)}^P$ of dropping it by the MLP model $\text{MLP}^P$ taking $[e_{t-1}^u + e_{t-1}^v, 1]$ as input, where $[\cdot, \cdot]$ denotes the concatenation operation. For any $(u,v)$ in $\overline{E}_{t-1}$, we compute the probability $p_{t,(u,v)}^P$ of adding an edge connecting $u$ and $v$ by $\text{MLP}^P$ taking $[e_{t-1}^u + e_{t-1}^v, 0]$ as input. Afterwards, for every $(u,v) \in E_{t-1}$, we randomly sample a binary value $o_{t,(u,v)}^P$ from the Bernoulli distribution parameterized with $p_{t,(u,v)}^P$, and drop $(u,v)$ from $E_{t-1}$ if $o_{t,(u,v)}^P = 1$. Similarly, we randomly sample $o_{t,(u,v)}^P$ for every $(u,v) \in \overline{E}_{t-1}$ but we will add $(u,v)$ into $E_{t-1}$ if $o_{t,(u,v)}^P = 1$.

An illustration of the process of producing $G_t$ from $G_{t-1}$ with our augmentation model is given in Figure 1. We also provide the detailed augmentation algorithm in Algorithm 1 of Appendix B.

### 3.4 LABEL-INVARIANCE OPTIMIZATION WITH REINFORCEMENT LEARNING

As our objective is generating label-invariant augmentations at most times, the ideal augmentation model $g$ should assign low transformation probabilities to graph elements corresponding to label-related information. For instance, when DropNode is used, if the dropping of some nodes will damage important graph substructures and cause label changing, the model $g$ should assign very low dropping probabilities to these nodes. However, we cannot directly make the model learn to produce label-invariant augmentations through supervised training because we do not have ground truth labels denoting which graph elements are important and should not be modified. To tackle this issue, we implicitly optimize the model with a reinforcement learning based training method.

We formulate the sequential graph augmentations as a Markov Decision Process (MDP). This is intuitive and reasonable, because the Markov property is naturally satisfied, i.e., the output graph at any transformation step is only dependent on the input graph, not on previously performed transformation. Specifically, at the $t$-th augmentation step, we define $G_{t-1}$, the graph obtained from the last step, as the current state, and the process of augmenting $G_{t-1}$ to $G_t$ is defined as state transition. The action is defined as the augmentation transformation $a_t$ generated from the model $g$, which includes the augmentation category $c_t$ and the exact transformations performed on all elements of $G_{t-1}$. The probability $p(a_t)$ of taking action $a_t$ for different $c_t$ is is described as below.

- If $c_t$ is MaskNF, then the transformation probability is the product of masking or unmasking probabilities for features of all nodes in $V_{t-1}$, so $p(a_t)$ is defined as

$$p(a_t) = p(c_t) * \prod_{v \in V_{t-1}} \prod_{k=1}^{d} \left(p_{t,v}^M[k]\right)^{o_{t,v}^M[k]} \left(1 - p_{t,v}^M[k]\right)^{1-o_{t,v}^M[k]} . \tag{4}$$

- If $c_t$ is DropNode, then the transformation probability is the product of dropping or non-dropping probabilities for all nodes in $V_{t-1}$, so $p(a_t)$ is defined as

$$p(a_t) = p(c_t) * \prod_{v \in V_{t-1}} \left(p_{t,v}^D\right)^{o_{t,v}^D} \left(1 - p_{t,v}^D\right)^{1-o_{t,v}^D} . \tag{5}$$

- If $c_t$ is PerturbEdge, then the transformation probability is the product of perturbing or non-perturbing probabilities for all edges in $E_{t-1}$ and $\overline{E}_{t-1}$, so $p(a_t)$ is defined as

$$p(a_t) = p(c_t) * \prod_{(u,v) \in E_{t-1} \cup \overline{E}_{t-1}} \left(p_{t,(u,v)}^P\right)^{o_{t,(u,v)}^P} \left(1 - p_{t,(u,v)}^P\right)^{1-o_{t,(u,v)}^P} . \tag{6}$$

We use the predicted label-invariance probabilities from a reward generation model $s$ as the feedback reward signal in the above reinforcement learning environment. We use graph matching network (Li et al., 2019) as the backbone of the reward generation model $s$ (see Appendix C for detailed introduction). When the sequential augmentation process starting from the graph $G_0$ ends, $s$ takes $(G_0, G_T)$ as inputs and outputs $s(G_0, G_T)$, which denotes the probability that the label is invariant after mapping the graph $G_0$ to the graph $G_T$. We use the logarithm of the predicted label-invariance probability, i.e., $R_T = \log s(G_0, G_T)$, as the return of the sequential augmentation process. Then the augmentation model $g$ is optimized by the REINFORCE algorithm (Sutton et al., 2000), which updates the model by the policy gradient $\hat{g}_\theta$ computed as $\hat{g}_\theta = R_T \nabla_\theta \sum_{t=1}^{T} \log p(a_t)$, where $\theta$ denotes the trainable parameters of $g$.

Prior to training the augmentation model $g$, we first train the reward generation model on manually sampled graph pairs from the training dataset. Specifically, a graph pair $(G_1, G_2)$ is first sampled from the dataset and passed into the reward generation model to predict the probability that $G_1$ and $G_2$ have the same label. Afterwards, the model is optimized by minimizing the binary cross entropy loss. During the training of the augmentation model $g$, the reward generation model is only used to generate rewards, so its parameters are fixed. See Algorithm 2 and 3 in Appendix B for the detailed training algorithm of reward generation model and augmentation model.

### 3.5 DISCUSSIONS AND RELATIONS WITH PRIOR METHODS

**Advantages of our method.** Our method explicitly estimates the transformation probability of each graph element by the automated augmentation model, thereby eliminating the negative effect of adopting a uniform transformation probability. Also, the reinforcement learning based training method can effectively help the model detect critical label-related information in the input graph, so the model can avoid damaging it and produce label-invariant augmentations with greater chances. We will show these advantages through extensive empirical studies in Section 4.1 and 4.2. Besides, the use of sequential augmentation, i.e., multiple steps of augmentation, can naturally help produce more diverse augmentations and samples, and the downstream classification model can benefit from diverse training samples. We will demonstrate it through ablation studies in Section 4.3.

**Relations with prior automated graph augmentations.** Several automated graph augmentation methods (You et al., 2021; Yin et al., 2022; Suresh et al., 2021; Hassani & Khasahmadi, 2022) have been proposed to generate multiple graph views for contrastive learning based pre-training. However, their augmentation models are optimized by contrastive learning objectives, which are not related to graph labels. Hence, their augmentation methods may still damage label-related information, and we experimentally show that they do not perform as well as GraphAug in supervised learning scenarios in Section 4.2. Though a recent study (Trivedi et al., 2022) claims that label-invariance is also important in contrastive learning, to our best knowledge, no automated graph augmentations have been proposed to preserve label-invariance in contrastive learning. Besides, we notice that a very recent study (Yue et al., 2022) also proposes a label-invariant automated augmentation method named GLA for semi-supervised graph classification. However, GLA is fundamentally

Table 1: The testing accuracy on the COLORS and TRIANGLES datasets with the GIN model. We report the average accuracy and standard deviation over ten runs on fixed train/validation/test splits.

| Dataset | No augmentation | Uniform MaskNF | Uniform DropNode | Uniform PerturbEdge | Uniform Mixture | GraphAug |
|---------|-----------------|----------------|------------------|---------------------|-----------------|----------|
| COLORS | 0.578±0.012 | 0.507±0.014 | 0.547±0.012 | 0.618±0.014 | 0.560±0.016 | **0.633±0.009** |
| TRIANGLES | 0.506±0.006 | 0.509±0.020 | 0.473±0.006 | 0.303±0.010 | 0.467±0.007 | **0.513±0.006** |

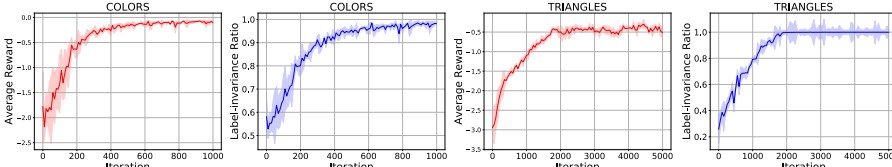

Figure 2: The changing curves of average rewards and label-invariance ratios on the validation set of the COLORS and TRIANGLES datasets as the augmentation model training proceeds. The results are averaged over ten runs, and the shadow shows the standard deviation.

different from GraphAug. For a graph data sample, GLA first obtains its graph-level representation by a GNN encoder. Then the augmentations are performed by perturbing the representation vector and label-invariant representations are selected by an auxiliary classification model. However, our GraphAug directly augments the graph data samples, and label-invariance is ensured by our proposed training method based on reinforcement learning. Hence, GraphAug can generate new data samples to enrich the existing training dataset while GLA cannot achieve it.

Due to the space limitation, we will discuss computational cost, augmentation step number, pretraining reward generation models, limitations, and relation with more prior methods in Appendix D.

## 4 EXPERIMENTS

In this section, we evaluate the proposed GraphAug method on two synthetic graph datasets and seven benchmark datasets. We show that in various graph classification tasks, GraphAug can consistently outperform previous graph augmentation methods. In addition, we conduct extensive ablation studies to evaluate the contributions of some components in GraphAug.

### 4.1 EXPERIMENTS ON SYNTHETIC GRAPH DATASETS

**Data.** We first show that our method can indeed produce label-invariant augmentations and outperform uniform transformations through experiments on two synthetic graph datasets COLORS and TRIANGLES, which are synthesized by running the open sourced data synthesis code[1] of Knyazev et al. (2019). The task of COLORS dataset is classifying graphs by the number of green nodes, and the color of a node is specified by its node feature. The task of TRIANGLES dataset is classifying graphs by the number of triangles (three-node cycles). We use fixed train/validation/test splits for experiments on both datasets. See more information about these two datasets in Appendix E.1.

**Setup.** We first train the reward generation model until it converges, then train the automated augmentation model. To check whether our augmentation model can learn to produce label-invariant augmentations, at different training iterations, we calculate the average rewards and the label-invariance ratio achieved after augmenting graphs in the validation set. Note that since we explicitly know how to obtain the labels of graphs from data generation codes, we can calculate label-invariance ratio, i.e., the ratio of augmented graphs that preserve their labels. To compare GraphAug with other augmentation methods, we train a GIN (Xu et al., 2019) based classification model with different augmentations for ten times, and report the averaged testing classification accuracy. We compare our GraphAug method with not using any data augmentations, and four graph augmentation baseline methods. Specifically, the augmentation methods using uniform MaskNF, DropNode, and PerturbEdge transformations, and a mixture of these three uniform transformations (Uniform Mixture), i.e., randomly picking one to augment graphs at each time, are used as baselines. To en-

[1]https://github.com/bknyaz/graph_attention_pool

Table 2: The performance on seven benchmark datasets with the GIN model. We report the average ROC-AUC and standard deviation over ten runs for the ogbg-molhiv dataset, and the average accuracy and standard deviations over three 10-fold cross-validation runs for the other datasets. Note that for JOAOv2, AD-GCL, and AutoGCL, we evaluate the augmentation methods of them under the supervised learning setting, so the numbers here are different from those in their papers.

| Method | PROTEINS | IMDB-BINARY | COLLAB | MUTAG | NCI109 | NCI1 | ogbg-molhiv |
|---|---|---|---|---|---|---|---|
| No augmentation | 0.704±0.004 | 0.731±0.004 | 0.806±0.003 | 0.827±0.013 | 0.794±0.003 | 0.804±0.003 | 0.756±0.014 |
| Uniform MaskNF | 0.702±0.008 | 0.720±0.006 | 0.815±0.002 | 0.788±0.012 | 0.777±0.006 | 0.794±0.002 | 0.741±0.010 |
| Uniform DropNode | 0.707±0.004 | 0.728±0.006 | 0.815±0.004 | 0.787±0.003 | 0.777±0.002 | 0.787±0.003 | 0.717±0.011 |
| Uniform PerturbEdge | 0.668±0.006 | 0.728±0.007 | 0.816±0.003 | 0.764±0.008 | 0.555±0.014 | 0.545±0.006 | 0.755±0.013 |
| Uniform Mixture | 0.707±0.004 | 0.730±0.009 | 0.815±0.003 | 0.779±0.014 | 0.776±0.006 | 0.783±0.003 | 0.746±0.010 |
| DropEdge | 0.707±0.002 | 0.733±0.012 | 0.812±0.003 | 0.779±0.005 | 0.762±0.007 | 0.780±0.002 | 0.762±0.010 |
| M-Mixup | 0.706±0.003 | 0.736±0.004 | 0.811±0.005 | 0.798±0.015 | 0.788±0.005 | 0.803±0.003 | 0.753±0.013 |
| $\mathcal{G}$-Mixup | 0.715±0.006 | 0.748±0.004 | 0.811±0.009 | 0.805±0.020 | 0.654±0.043 | 0.686±0.037 | 0.771±0.005 |
| FLAG | 0.709±0.007 | 0.747±0.008 | 0.803±0.006 | 0.835±0.015 | 0.804±0.002 | 0.804±0.002 | 0.765±0.011 |
| JOAOv2 | 0.700±0.003 | 0.707±0.008 | 0.688±0.003 | 0.775±0.016 | 0.675±0.003 | 0.670±0.006 | 0.744±0.014 |
| AD-GCL | 0.699±0.008 | 0.712±0.008 | 0.670±0.008 | 0.837±0.010 | 0.634±0.003 | 0.641±0.004 | 0.762±0.013 |
| AutoGCL | 0.684±0.008 | 0.707±0.007 | 0.745±0.002 | 0.783±0.022 | 0.705±0.003 | 0.737±0.002 | 0.704±0.016 |
| GraphAug | **0.722±0.004** | **0.762±0.004** | **0.829±0.002** | **0.853±0.008** | **0.811±0.002** | **0.816±0.001** | **0.774±0.010** |

sure fair comparison, we use the same hyper-parameter setting in training classification models for all methods. See hyper-parameters and more experimental details in Appendix E.1.

**Results.** The changing curves of average rewards and label-invariance ratios are visualized in Figure 2. These curves show that as the training proceeds, our model can gradually learn to obtain higher rewards and produce augmentations leading to higher label-invariance ratio. In other words, they demonstrate that our augmentation model can indeed learn to produce label-invariant augmentations after training. The testing accuracy of all methods on two datasets are presented in Table 1. From the results, we can clearly find using some uniform transformations that do not satisfy label-invariance, such as uniform MaskNF on the COLORS dataset, achieve much worse performance than not using augmentations. However, using augmentations produced by the trained GraphAug models can consistently achieve the best performance, which demonstrates the significance of label-invariant augmentations to improving the performance of graph classification models. We further study the training stability and generalization ability of GraphAug models, conduct an exploration experiment about training GraphAug models with adversarial learning, compare with some manually designed label-invariant augmentations, and compare label-invariance ratios with baseline methods on the COLORS and TRIANGLES datasets. See Appendix F.1, F.2, F.3, and F.6 for details.

## 4.2 EXPERIMENTS ON GRAPH BENCHMARK DATASETS

**Data.** We further demonstrate the advantages of our GraphAug method over previous graph augmentation methods on six widely used datasets from the TUDatasets benchmark (Morris et al., 2020), including MUTAG, NCI109, NCI1, PROTEINS, IMDB-BINARY, and COLLAB. We also conduct experiments on the ogbg-molhiv dataset, which is a large molecular graph dataset from the OGB benchmark (Hu et al., 2020). See more information about datasets in Appendix E.2.

**Setup.** We evaluate the performance by testing accuracy for the six datasets of the TUDatasets benchmark, and use testing ROC-AUC for the ogbg-molhiv dataset. We use two classification models, including GIN (Xu et al., 2019) and GCN (Kipf & Welling, 2017). We use the 10-fold cross-validation scheme with train/validation/test splitting ratios of 80%/10%/10% on the datasets from the TU-Datasets benchmark, and report the averaged testing accuracy over three different runs. For the ogbg-molhiv dataset, we use the official train/validation/test splits and report the averaged testing ROC-AUC over ten runs. In addition to the baselines in Section 4.1, we also compare with previous graph augmentation methods, including DropEdge (Rong et al., 2020), M-Mixup (Wang et al., 2021), $\mathcal{G}$-Mixup (Han et al., 2022), and FLAG (Kong et al., 2022). Besides, we compare with three automated augmentations proposed for graph self-supervised learning, including JOAOv2 (You et al., 2021), AD-GCL (Suresh et al., 2021), and AutoGCL (Yin et al., 2022). Note that we take their trained augmentation modules as the data augmenter, and evaluate the performance of supervised classification models trained on the samples produced by these data augmenters. For fair comparison, we use the same hyper-parameter setting in training classification models for GraphAug and baseline methods. See hyper-parameters and more experimental details in Appendix E.2.

Table 3: Results of ablation studies about learnable and sequential augmentation. We report the average accuracy and standard deviation over three 10-fold cross-validation runs with the GIN model.

| Method | PROTEINS | IMDB-BINARY | NCI1 |
|---|---|---|---|
| GraphAug w/o learnable graph transformation | 0.696±0.006 | 0.724±0.003 | 0.760±0.003 |
| GraphAug w/o learnable category selection | 0.702±0.004 | 0.746±0.009 | 0.796±0.006 |
| GraphAug w/o sequential augmentation | 0.712±0.002 | 0.753±0.003 | 0.809±0.004 |
| GraphAug | **0.722±0.004** | **0.762±0.004** | **0.816±0.001** |

**Results.** The performance of different methods on all seven datasets with the GIN model is summarized in Table 2, and see Table 9 in Appendix F.4 for the results of the GCN model. According to the results, our GraphAug method can achieve the best performance among all graph augmentation methods over seven datasets. Similar to the results in Table 1, for molecule datasets including MUTAG, NCI109, NCI1, and ogbg-molhiv, using some uniform transformations based augmentation methods dramatically degrades the classification accuracy. On the other hand, our GraphAug method consistently outperforms baseline methods, such as mixup methods and existing automated data augmentations in graph self-supervised learning. The success on graph benchmark datasets once again validates the effectiveness of our proposed GraphAug method.

## 4.3 ABLATION STUDIES

In addition to demonstrating the effectiveness of GraphAug, we conduct a series of ablation experiments and use empirical results to answer (1) why we make augmentation automated and learnable, (2) why we use sequential, multi-step augmentation, (3) why we adopt a combination of three different transformations (MaskNF, DropNode, PerturbEdge) instead of using only one, and (4) why we use virtual nodes. We present the ablation studies (1) and (2) in this section and leave (3) and (4) in Appendix F.5. For all ablation studies, we train GIN based classification models on the PROTEINS, IMDB-BINARY, and NCI1 datasets, and use the same evaluation pipeline as Section 4.2.

**Ablation on learnable graph transformation and category selection.** We first show that making the model learn to generate graph transformations for each graph element and select augmentation category are both important. We compare with a variant of GraphAug that does not learn graph transformations but simply adopts uniform transformations, and another variant that randomly select the category of graph transformation, instead of explicitly predicting it. The classification accuracy on three datasets of these two variants are presented in the first two rows of Table 3. Results show that the performance of two variants is worse, and particularly removing learnable graph transformation will significantly degrade the performance. It is demonstrated that learning to generate graph transformations and select augmentation category are both key success factors of GraphAug.

**Ablation on sequential augmentation.** We next show the advantage of sequential augmentation over one-step augmentation. We compare with the variant of GraphAug that performs only one step of augmentation, i.e., with the augmentation step number T=1, and present its performance in the third row of Table 3. It is clear that using one step of augmentation will result in worse performance over all datasets. We think this demonstrates that the downstream classification model will benefit from the diverse training samples generated from sequential and multi-step augmentation.

## 5 CONCLUSIONS AND FUTURE WORK

We propose GraphAug, the first automated data augmentation framework for graph classification. GraphAug considers graph augmentations as a sequential transformation process. To eliminate the negative effect of uniform transformations, GraphAug uses an automated augmentation model to generate transformations for each element in the graph. In addition, GraphAug adopts a reinforcement learning based training procedure, which helps the augmentation model learn to avoid damaging label-related information and produce label-invariant augmentations. Through extensive empiric studies, we demonstrate that GraphAug can achieve better performance than many existing graph augmentation methods on various graph classification tasks. In the future, we would like to explore simplifying the current training procedure of GraphAug and applying GraphAug to other graph representation learning problems, such as the node classification problem.

## REPRODUCIBILITY STATEMENT

We have provided the detailed algorithm pseudocodes in Appendix B and experimental setting details in Appendix E for reproducing the results. The source codes of our method are included in DIG (Liu et al., 2021) library.

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

# A  VISUALIZATION OF DIFFERENT AUGMENTATION METHODS

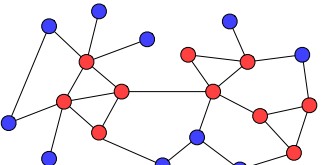 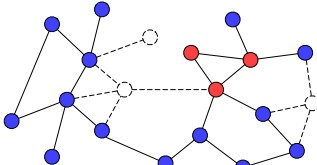 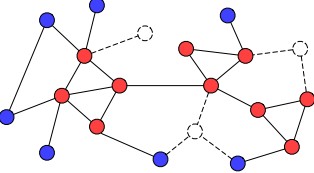

(a) An illustration of a data sample from the TRIANGLES dataset. Red nodes represent the nodes belonging to triangles. The label of this data sample is 4 since there are four triangles. Training without any augmentations on the TRIANGLES dataset achieves the average testing accuracy of $0.506 \pm 0.006$.

(b) The data sample generated by augmenting the data sample in (a) with the uniform DropNode transformation. Note that two nodes originally belonging to triangles are removed, and the label is changed to 1. Training with the uniform DropNode transformation achieves the average testing accuracy of $0.473 \pm 0.006$.

(c) The data sample generated by augmenting the data sample in (a) with the label-invariant DropNode transformation (the DropNode with GT method in Appendix F.3), which intentionally avoids dropping nodes in triangles. Training with this label-invariant augmentation achieves the average testing accuracy of $0.522 \pm 0.007$.

Figure 3: Comparison of different augmentation methods on the TRIANGLES dataset. We use a GIN (Xu et al., 2019) based classification model to evaluate different augmentation methods, and report the average accuracy and standard deviation over ten runs on a fixed train/validation/test split. In (a), we show a graph data sample with 4 triangles. In (b) and (c), we the data samples generated by augmenting the data sample in (a) with two different augmentation methods. We can clearly find that using the uniform DropNode transformation degrades the classification performance but using the label-invariant augmentation improves the performance.

# B   AUGMENTATION AND TRAINING ALGORITHMS

---

**Algorithm 1:** Augmentation Algorithm of GraphAug

---

1: **Input:** Graph $G_0 = (V_0, E_0, X_0)$; total number of augmentation steps $T$; the augmentation
   model $g$ composed of GNN-encoder, GRU, and four MLP models $\text{MLP}^C$, $\text{MLP}^M$, $\text{MLP}^D$,
   $\text{MLP}^P$
2: Initialize the hidden state $q_0$ of the GRU model to zero vector
3: **for** $t = 1$ **to** $T$ **do**
4:    Obtain $G'_{t-1}$ by adding a virtual node to $G_{t-1}$
5:    $e_{t-1}^{virtual}, \{e_{t-1}^v : v \in V_{t-1}\} = \text{GNN-encoder}(G'_{t-1})$
6:    $q_t = \text{GRU}(q_{t-1}, e_{t-1}^{virtual})$
7:    $p_t^C = \text{MLP}^C(q_t)$
8:    Sample $c_t$ from the categorical distribution of $p_t^C$
9:    **if** $c_t$ is MaskNF **then**
10:       **for** $v \in V_{t-1}$ **do**
11:          $p_{t,v}^M = \text{MLP}^M(e_{t-1}^v)$
12:          Sample $o_{t,v}^M$ from the Bernoulli distribution parameterized with $p_{t,v}^M$
13:          **for** $k = 1$ **to** $d$ **do**
14:             Set the $k$-th node feature of $v$ to zero if $o_{t,v}^M[k] = 1$
15:    **else if** $c_t$ is DropNode **then**
16:       **for** $v \in V_{t-1}$ **do**
17:          $p_{t,v}^D = \text{MLP}^D(e_{t-1}^v)$
18:          Sample $o_{t,v}^D$ from the Bernoulli distribution parameterized with $p_{t,v}^D$
19:          Drop the node $v$ from $V_{t-1}$ if $o_{t,v}^D = 1$
20:    **else if** $c_t$ is PerturbEdge **then**
21:       Obtain the addable edge set $\overline{E}_{t-1}$ by randomly sampling at most $|E_{t-1}|$ addable edges
          from $\{(u, v) : u, v \in V_{t-1}, (u, v) \notin E_{t-1}\}$
22:       **for** $(u, v) \in E_{t-1}$ **do**
23:          $p_{t,(u,v)}^P = \text{MLP}^P\left([e_{t-1}^u + e_{t-1}^v, 1]\right)$
24:          Sample $o_{t,(u,v)}^P$ from the Bernoulli distribution parameterized with $p_{t,(u,v)}^P$
25:          Drop $(u, v)$ from $E_{t-1}$ if $o_{t,(u,v)}^P = 1$
26:       **for** $(u, v) \in \overline{E}_{t-1}$ **do**
27:          $p_{t,(u,v)}^P = \text{MLP}^P\left([e_{t-1}^u + e_{t-1}^v, 0]\right)$
28:          Sample $o_{t,(u,v)}^P$ from the Bernoulli distribution parameterized with $p_{t,(u,v)}^P$
29:          Add $(u, v)$ into $E_{t-1}$ if $o_{t,(u,v)}^P = 1$
30:    Set $G_t$ as the outputted graph from the $t$-th augmentation step
31: Output $G_T$

---

**Algorithm 2:** Training Algorithm of the reward generation model of GraphAug

---

1: **Input:** Graph dataset $\mathcal{D}$; batch size $B$; learning rate $\alpha$; the reward generation model $s$ with the
   parameter $\varphi$
2: **repeat**
3:    Sample a batch $\mathcal{G}$ of $B$ data samples from $\mathcal{D}$
4:    $L = 0$
5:    **for** $G \in \mathcal{G}$ **do**
6:       Randomly sample a graph $G^+$ with the same label as $G$ from $\mathcal{G}$ and a graph $G^-$ with
          different label as $G$
7:       $L = L - \log s(G, G^+) - \log(1 - s(G, G^-))$
8:    Update the parameter $\varphi$ of $s$ as $\varphi = \varphi - \alpha \nabla_\varphi L / B$
9: **until** the training converges
10: Output the trained reward generation model $s$

---

---

**Algorithm 3:** Training Algorithm of the augmentation model of GraphAug

---

1: **Input:** Graph dataset $\mathcal{D}$; batch size $B$; learning rate $\alpha$; total number of augmentation steps $T$; the augmentation model $g$ with the parameter $\theta$ composed of GNN-encoder, GRU, and four MLP models $\text{MLP}^C$, $\text{MLP}^M$, $\text{MLP}^D$, $\text{MLP}^P$; the trained reward generation model $s$

2: **repeat**

3:    Sample a batch $\mathcal{G}$ of $B$ data samples from $\mathcal{D}$

4:    $\hat{g}_\theta = 0$

5:    **for** $G \in \mathcal{G}$ **do**

6:        Set $G_0 = (V_0, E_0, X_0)$ as $G$

7:        Initialize the hidden state $q_0$ of the GRU model to zero vector

8:        **for** $t = 1$ **to** $T$ **do**

9:            Obtain $G'_{t-1}$ by adding a virtual node to $G_{t-1}$

10:           $e_{t-1}^{virtual}, \{e_{t-1}^v : v \in V_{t-1}\} = \text{GNN-encoder}(G'_{t-1})$

11:           $q_t = \text{GRU}(q_{t-1}, e_{t-1}^{virtual})$

12:           $p_t^C = \text{MLP}^C(q_t)$

13:           Sample $c_t$ from the categorical distribution of $p_t^C$, set $\log p(a_t) = \log p_t^C(c_t)$

14:           **if** $c_t$ is MaskNF **then**

15:               **for** $v \in V_{t-1}$ **do**

16:                   $p_{t,v}^M = \text{MLP}^M(e_{t-1}^v)$

17:                   Sample $o_{t,v}^M$ from the Bernoulli distribution parameterized with $p_{t,v}^M$

18:                   **for** $k = 1$ **to** $d$ **do**

19:                      $\log p(a_t) = \log p(a_t) + o_{t,v}^M[k] \log p_{t,v}^M[k] + (1 - o_{t,v}^M[k]) \log(1 - p_{t,v}^M[k])$

20:                      Set the $k$-th node feature of $v$ to zero if $o_{t,v}^M[k] = 1$

21:           **else if** $c_t$ is DropNode **then**

22:               **for** $v \in V_{t-1}$ **do**

23:                   $p_{t,v}^D = \text{MLP}^D(e_{t-1}^v)$

24:                   Sample $o_{t,v}^D$ from the Bernoulli distribution parameterized with $p_{t,v}^D$

25:                   $\log p(a_t) = \log p(a_t) + o_{t,v}^D \log p_{t,v}^D + (1 - o_{t,v}^D) \log(1 - p_{t,v}^D)$

26:                   Drop the node $v$ from $V_{t-1}$ if $o_{t,v}^D = 1$

27:           **else if** $c_t$ is PerturbEdge **then**

28:               Obtain the addable edge set $\overline{E}_{t-1}$ by randomly sampling at most $|E_{t-1}|$ addable edges from $\{(u,v) : u,v \in V_{t-1}, (u,v) \notin E_{t-1}\}$

29:               **for** $(u,v) \in E_{t-1}$ **do**

30:                   $p_{t,(u,v)}^P = \text{MLP}^P\left([e_{t-1}^u + e_{t-1}^v, 1]\right)$

31:                   Sample $o_{t,(u,v)}^P$ from the Bernoulli distribution parameterized with $p_{t,(u,v)}^P$

32:                   $\log p(a_t) = \log p(a_t) + o_{t,(u,v)}^P \log p_{t,(u,v)}^P + (1 - o_{t,(u,v)}^P) \log(1 - p_{t,(u,v)}^P)$

33:                   Drop $(u,v)$ from $E_{t-1}$ if $o_{t,(u,v)}^P = 1$

34:               **for** $(u,v) \in \overline{E}_{t-1}$ **do**

35:                   $p_{t,(u,v)}^P = \text{MLP}^P\left([e_{t-1}^u + e_{t-1}^v, 0]\right)$

36:                   Sample $o_{t,(u,v)}^P$ from the Bernoulli distribution parameterized with $p_{t,(u,v)}^P$

37:                   $\log p(a_t) = \log p(a_t) + o_{t,(u,v)}^P \log p_{t,(u,v)}^P + (1 - o_{t,(u,v)}^P) \log(1 - p_{t,(u,v)}^P)$

38:                   Add $(u,v)$ into $E_{t-1}$ if $o_{t,(u,v)}^P = 1$

39:            Set $G_t$ as the outputted graph from the $t$-th augmentation step

40:        $\hat{g}_\theta = \hat{g}_\theta + \log s(G_0, G_T)\nabla_\theta \sum_{t=1}^T \log p(a_t)$

41:    Update the parameter $\theta$ of $g$ as $\theta = \theta + \alpha\hat{g}_\theta/B$.

42: **until** the training converges

43: Output the trained augmentation model $g$

---

## C    DETAILS OF REWARD GENERATION MODEL

We use the graph matching network (Li et al., 2019) as the reward generation model $s$ to predict the probability $s(G_0, G_T)$ that $G_0$ and $G_T$ have the same label (here $G_0$ is a graph sampled from the dataset, i.e., the starting graph of the sequential augmentation process, and $G_T$ is the graph produced from $T$ steps of augmentation by the augmentation model). The graph matching network takes both $G_0 = (V_0, E_0, X_0)$ and $G_T = (V_T, E_T, X_T)$ as input, performs multiple message operations on them with a shared GNN model separately. The computational process of the message passing for any node $v$ in $G_0$ at the $\ell$-th layer of the model is

$$h_v^\ell = \text{UPDATE}\left(h_v^{\ell-1}, \text{AGG}\left(\left\{m_{jv}^\ell : j \in \mathcal{N}(v)\right\}\right), \mu_v^{G_T}\right), \tag{7}$$

which is the same as the message passing of vanilla GNNs in Equation (1) other than involving propagating the message $\mu_v^{G_T}$ from the graph $G_T$ to the node $v$ in $G_0$. The message $\mu_v^{G_T}$ is extracted by an attention based module as

$$w_{iv} = \frac{\exp\left(\text{sim}\left(h_v^{\ell-1}, h_i^{\ell-1}\right)\right)}{\sum_{u \in V_T} \exp\left(\text{sim}\left(h_v^{\ell-1}, h_u^{\ell-1}\right)\right)}, \mu_v^{G_T} = \sum_{i \in V_T} w_{iv}(h_v^{\ell-1} - h_i^{\ell-1}), v \in V_0, \tag{8}$$

where $\text{sim}(\cdot, \cdot)$ computes the similarity between two vectors by dot-product. The message passing for any node in $G_T$ is similarly computed as in Equation (7), and this also involves propagating message from $G_0$ to nodes in $G_T$ with the attention module in Equation (8). Afterwards, the graph-level representations $h_{G_0}$ and $h_{G_T}$ of $G_0$ and $G_T$ are separately obtained from their node embeddings as in Equation (2). We pass $|h_{G_0} - h_{G_T}|$, the element-wise absolute deviation of $h_{G_0}$ and $h_{G_T}$, to an MLP model to compute $s(G_0, G_T)$.

# D    More Discussions

**Discussions about computational cost.** Considering a graph with $|V|$ nodes and $|E|$ edges, the time complexity of performing our defined DropNode or MaskNF transformation on it is $O(|V|)$, and the time complexity is $O(|E|)$ for the PerturbEdge transformation since both edge dropping or addition is operated on at most $|E|$ edges. Hence, the time complexity of augmenting the graph for $T$ steps is $O(T|V| + T|E|)$. This time cost is affordable for most real-world applications. We test the average time used to augment a graph on each benchmark dataset used in our experiments with our trained augmentation model, see Table 10 for time results. We can find that for most dataset, our method only takes a very small amount of time ($< 0.05$ s) to augment a graph in average. Besides, during the training of the augmentation model, the computation of rewards by the reward generation model involves attention module (see Equation (9)), which causes an extra computational cost of $O(|V|^2)$. In practice, this does not have much effect on small graphs, but may lead to large computation and memory cost on large graphs.

**Discussions about augmentation step number.** For the number of augmentation steps $T$, we do not let the model to learn or decide $T$ itself but make $T$ a fixed hyper-parameter to avoid the model being stuck in the naive solution of not doing augmentation at all (i.e., learn $T = 0$). This strategy is also adopted by previous image augmentation method (e.g. AutoAugment (Cubuk et al., 2019)). A larger $T$ encourages the model to produce more diverse augmentations but makes it harder to keep label-invariance. We experimentally find that if $T \geq 8$, it is hard to obtain sufficiently high reward for the model. Hence, we tune $T$ in $[1, 8]$ for each dataset to achieve the best trade-off between producing diverse augmentations and keeping label-invariance.

**Discussions about pre-training reward generation model.** In our method, before training the augmentation model, we first pre-train the reward generation model and make it fixed while training the augmentation model. Such a training pipeline has both advantages and disadvantages. The advantages of using the fixed/pre-trained reward generation model are two-fold. (1) First, pre-training the reward generation model enables it to accurately predict whether two input graphs have the same labels or not, so that the generated reward signals can provide accurate feedback for the augmentation model. (2) Second, using the fixed reward generation model can stabilize the training of the augmentation model in practice. As we shown in Appendix F.2, if the reward generation model is not fixed and jointly trained with the augmentation model, the training becomes unstable and models consistently diverge. The disadvantage of pre-training the reward generation model is that this training pipeline is time-consuming, because we have to train two models every time to obtain the finally usable graph augmenter.

**Limitations of our method.** There are some limitations in our method. (1) First, our method adopts a complicated two-step training pipeline which first trains the reward generation model and then trains the augmentation model. We have tried simplifying it to one-step training through adversarial training as in Ratner et al. (2017b). However, we found it to be very unstable and the augmentation model consistently diverges (see Appendix F.2 for an exploration experiment about adversarial training on the COLORS and TRIANGLES dataset). We leave the problem of simplifying the training to the future. (2) Second, our augmentation method will take extra computational cost in both training the augmentation model and providing augmented samples for the downstream graph classification training. The time and resource cost of training models can be large on the large datasets. For instance, on the ogbg-molhiv dataset, we find it takes the total time of around 10 hours to train the reward generation model and augmentation model before we obtain a finally usable graph augmenter. Given that the performance improvement is not significant on the ogbg-molhiv dataset, such a large time cost is not a worthwhile investment. **Our GraphAug mainly targets on improving the graph classification performance by generating more training data samples for the tasks with small datasets, particularly for those that need huge cost to manually collect and label new data samples.** But for the classification task with sufficient training data, the benefits of using GraphAug are limited and not worth the large time and resource cost to train GraphAug models.

**Relations with automated image augmentations.** GraphAug are somehow similar to some automated image augmentations (Cubuk et al., 2019; Zhang et al., 2020) in that they both use sequential augmentation and reinforcement learning based training. However, they are actually fundamentally different. Label-invariance is not a problem in automated image augmentations because the used image transformations ensure label-invariance. On the other hand, as discussed in Section 3.2,

it is non-trivial to make graph transformations ensure label-invariance. In GraphAug, the learnable graph transformation model and the reinforcement learning based training are used to produce label-invariant augmentations, which are actually the main contribution of GraphAug. Another fundamental difference between GraphAug and automated image augmentations lies in the reward design. Many automated image augmentation methods, such as AutoAugment (Cubuk et al., 2019), train a child network model on the training data and use the achieved classification accuracy on the validation data as the reward. Instead, our GraphAug uses the label-invariance probability predicted by the reward generation model as the reward signal to train the augmentation model. We argue that such reward design has several advantages over using the classification accuracy as the reward. (1) First, maximizing the label-invariance probability can directly encourage the augmentation model to produce label-invariant augmentations. However, the classification accuracy is not directly related to label-invariance, so using it as the reward feedback does not necessarily make the augmentation model learn to ensure label-invariance. (2) Second, predicting the label-invariance probability only needs one simple model inference process that is computationally cheap, while obtaining the classification accuracy is computationally expensive because it needs to train a model from scratch. (3) Most importantly, our reward generation scheme facilitates the learning of the augmentation model by **providing the reward feedback for every individual graph**. Even in the same dataset, the label-related structures or patterns in different graphs may vary a lot, hence, good augmentation strategies for different graphs can be different. However, the classification accuracy evaluates the classification performance when using the produced graph augmentations to train models on the overall dataset, which does not provide any feedback about whether the produced augmentation on every individual graph sample is good or not. Differently, the label-invariance probability is computed for every individual graph sample, thereby enabling the model to capture good augmentation strategies for every individual graph. Considering these advantages, we do not use the classification accuracy but take the label-invariance probability predicted by the reward generation model as the reward. Overall, **GraphAug cannot be considered as a simple extension of automated image augmentations to graphs.**

**Relation with prior graph augmentation methods.** In addition to GLA (Yue et al., 2022), we also notice Graphair (Ling et al., 2023), another recently proposed automated graph augmentation method. However, Graphair aims to produce fairness-aware graphs for fair graph representation learning, while our method is proposed for graph classification. Additionally, graph mixup methods (Wang et al., 2021; Han et al., 2022; Guo & Mao, 2021; Park et al., 2022) synthesize a new graph or graph representation from two input graphs. Because the new data sample is assigned with the combination of labels of two input graphs, mixup operations are supposed to detect and mix the label-related information of two graphs (Guo & Mao, 2021). However, our method is simpler and more intuitive because it only needs to detect and preserve the label-related information of one input graph. In addition, another method FLAG (Kong et al., 2022) can only augment node features, while our method can produce augmentations in node features, nodes and edges. Besides, similar to the motivation of our GraphAug, some other studies have also found that preserving important structures or node features is significant in designing effective graph augmentations. A pioneering method in this direction is GCA (Zhu et al., 2021b), which proposes to identify important edges and node features in the graph by node centralities. GCA augments the graph by random edge dropping and node feature masking, but assigns lower perturbation probabilities to the identified important edges and node features. Also, other studies (Wang et al., 2020; Bicciato & Torsello, 2022; Zhou et al., 2020b) assume that some motif or subgraph structures in the graph is significant, and propose to augment graphs by manually designed transformations to avoid removing them. Overall, these augmentations are based on some rules or assumptions about how to preserve important structures of the input graph. Differently, our GraphAug method does not aim to define a fixed graph augmentation strategy for every graph. Instead, it seeks to make the augmentation model find good augmentation strategies automatically with reinforcement learning based training.

**Relations with graph explainability.** Our method is related to graph explainability in that the predicted transformation probabilities from our augmentation model $g$ is similar to explainability scores of some graph explainability methods (Maruhashi et al., 2018; Yuan et al., 2020; 2021). Hence, we hope that our augmentation method can bring inspiration to researchers in the graph explainability area.

Table 4: Statistics of graph benchmark datasets.

| Datasets | # graphs | Average # nodes | Average # edges | # classes |
|---|---|---|---|---|
| PROTEINS | 1113 | 39.06 | 72.82 | 2 |
| IMDB-BINARY | 1000 | 19.77 | 96.53 | 2 |
| COLLAB | 5000 | 74.49 | 2457.78 | 3 |
| MUTAG | 188 | 17.93 | 19.79 | 2 |
| NCI109 | 4127 | 29.68 | 32.13 | 2 |
| NCI1 | 4110 | 29.87 | 32.30 | 2 |
| ogbg-molhiv | 41,127 | 25.5 | 27.5 | 2 |

Table 5: Some hyper-parameters for the reward generation model and its training.

| Datasets | # layers | batch size | # training epochs |
|---|---|---|---|
| PROTEINS | 6 | 32 | 420 |
| IMDB-BINARY | 6 | 32 | 320 |
| COLLAB | 5 | 8 | 120 |
| MUTAG | 5 | 32 | 230 |
| NCI109 | 5 | 32 | 200 |
| NCI1 | 5 | 32 | 200 |
| ogbg-molhiv | 5 | 32 | 200 |

# E   MORE DETAILS ABOUT EXPERIMENTAL SETTING

## E.1   EXPERIMENTS ON SYNTHETIC GRAPH DATASETS

**Data information.** We synthesize the COLORS and TRIANGLES dataset by running the open sourced data synthesis code of Knyazev et al. (2019). For the COLORS dataset, we synthesize 8000 graphs for training, 1000 graphs for validation, and 1000 graphs for testing. For the TRIANGLES dataset, we synthesize 30000 graphs for training, 5000 graphs for validation, and 5000 graphs for testing. The labels of all data samples in both datasets belong to $\{1, ..., 10\}$.

**Details of the model and training.** The Adam optimizer (Kingma & Ba, 2015) is used for the training of all models. For both datasets, we use a reward generation model with 5 layers and the hidden size of 256, and the graph level embedding is obtained by sum pooling. It is trained for 1 epoch on the COLORS dataset and 200 epochs on the TRIANGLES dataset. The batch size is 32 and the learning rate is 0.0001. For the augmentation model, we use a GIN model with 3 layers and the hidden size of 64 for GNN encoder, an MLP model with 2 layers, the hidden size of 64, and ReLU as the non-linear activation function for $MLP^C$, and an MLP model with 2 layers, the hidden size of 128, and ReLU as the non-linear activation function for $MLP^M$, $MLP^D$, and $MLP^P$. The augmentation model is trained for 5 epochs with the batch size of 32 and the learning rate of 0.0001 on both datasets. To stabilize the training of the augmentation model, we manually control the augmentation model to only modify 5% of graph elements at each augmentation step during the training. On the COLORS dataset, we use a classification model where the number of layers is 3, the hidden size is 128, and the readout layer is max pooling. On the TRIANGLES dataset, we use a classification model where the number of layers is 3, the hidden size is 64, and the readout layer is sum pooling. On both datasets, we set the training batch size as 32 and the learning rate as 0.001 when training classification models, and all classification models are trained for 100 epochs.

## E.2   EXPERIMENTS ON GRAPH BENCHMARK DATASETS

**Data information.** We use six datasets from the TUDatasets benchmark (Morris et al., 2020), including three molecule datasets MUTAG, NCI109, NCI1, one bioinformatics dataset PROTEINS, and two social network datasets IMDB-BINARY and COLLAB. We also use the ogbg-molhiv

Table 6: Some hyper-parameters for the augmentation model and its training.

| Datasets | # augmentation steps $T$ | batch size | # training epochs |
|---|---|---|---|
| PROTEINS | 2 | 32 | 30 |
| IMDB-BINARY | 8 | 32 | 30 |
| COLLAB | 8 | 32 | 10 |
| MUTAG | 4 | 16 | 200 |
| NCI109 | 2 | 32 | 20 |
| NCI1 | 2 | 32 | 20 |
| ogbg-molhiv | 2 | 128 | 10 |

Table 7: Some hyper-parameters for the classification model and its training.

| Datasets | # layers | hidden size | batch size |
|---|---|---|---|
| PROTEINS | 3 | 128 | 32 |
| IMDB-BINARY | 4 | 128 | 32 |
| COLLAB | 4 | 64 | 32 |
| MUTAG | 4 | 128 | 16 |
| NCI109 | 4 | 128 | 32 |
| NCI1 | 3 | 128 | 32 |
| ogbg-molhiv | 5 | 300 | 32 |

dataset from the OGB benchmark (Hu et al., 2020). See Table 4 for the detailed statistics of all benchmark datasets used in our experiments.

**Details of model and training.** The Adam optimizer (Kingma & Ba, 2015) is used for training of all models. For all six datasets, we set the hidden size as 256 and the readout layer as sum pooling for the reward generation model, and the reward generation model is trained using 0.0001 as the learning rate. See other hyper-parameters about the reward generation model and its training in Table 5. The hyper-parameters of the augmentation model is the same as those in experiments of synthetic graph datasets and the learning rate is 0.0001 during its training, but we tune the batch size, the training epochs and the number of augmentation steps $T$ on each dataset. See Table 6 for the optimal values of them on each dataset. The strategy of modifying only 5% of graph elements is also used during the training of augmentation models. Besides, for classification models, we set the readout layer as mean pooling, and tune the number of layers, the hidden size, and the training batch size. See Table 7 for these hyper-parameters. All classification models are trained for 100 epochs with the learning rate of 0.001.

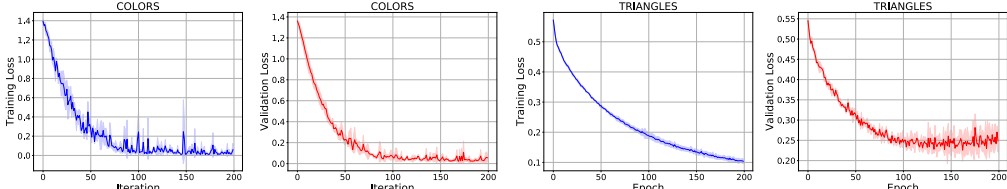

Figure 4: The changing curves of training and validation loss on the COLORS and TRIANGLES datasets when training the reward generation model of GraphAug with Algorithm 2. The results are averaged over ten runs, and the shadow shows the standard deviation.

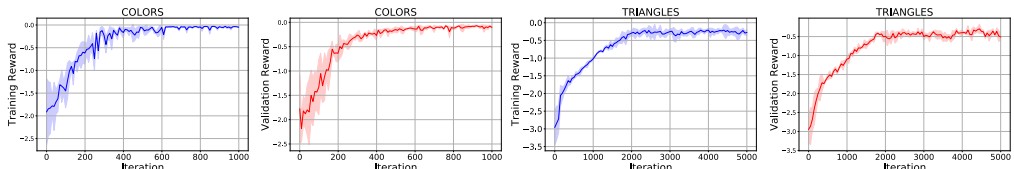

Figure 5: The changing curves of training and validation rewards on the COLORS and TRIANGLES datasets when training the augmentation model of GraphAug with Algorithm 3. The results are averaged over ten runs, and the shadow shows the standard deviation.

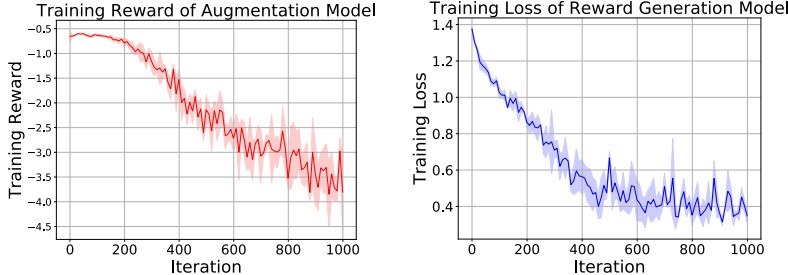

Figure 6: The changing curves of training rewards of augmentation model and training loss of reward generation model when training two models together with adversarial learning on the COLORS dataset. The results are averaged over ten runs, and the shadow shows the standard deviation.

## F    MORE EXPERIMENTAL RESULTS

### F.1    STUDY OF TRAINING STABILITY AND GENERALIZATION

Taking the COLORS and TRIANGLES datasets as examples, we show the learning curves of reward generation models and augmentation models in Figure 4 and Figure 5, respectively. The learning curves on the training set show that the training is generally very stable for both reward generation models and augmentation models since no sharp oscillation happens. Comparing the learning curves on the training and validation set, we can find that on the COLORS dataset, the curves converge to around the same loss and rewards on the training and validation set when the training converges. Hence, reward generation model and the augmentation model both have very good generalization abilities. Differently, on more complicated TRIANGLES dataset, slight overfitting exists for both models but the overall generalization ability is still acceptable. Actually, to eliminatee the negative effect of overfitting, we always take the reward generation model with lowest validation loss and the augmentation model with highest validation reward in our experiments. In a word, our studies about training stability and generalization show that both the reward generation model and augmentation model can be trained stably and have acceptable generalization ability.

Table 8: The testing accuracy on the COLORS and TRIANGLES datasets with the GIN model. We report the average accuracy and standard deviation over ten runs on fixed train/validation/test splits.

| Dataset | No augmentation | MaskNF with GT | DropNode with GT | PerturbEdge with GT | GraphAug |
|---------|-----------------|----------------|------------------|---------------------|----------|
| COLORS | 0.578±0.012 | 0.627±0.013 | 0.627±0.017 | n/a | 0.633±0.009 |
| TRIANGLES | 0.506±0.006 | n/a | 0.522±0.007 | 0.524±0.006 | 0.513±0.006 |

## F.2 ADVERSARIAL TRAINING EXPERIMENT

One possible one-stage training alternative of GraphAug is the adversarial training strategy in Ratner et al. (2017b). Specifically, the augmentation model is trained jointly with the reward generation model. We construct the positive graph pair $(G, G^+)$ sampled from the dataset in which $G$ and $G^+$ have the same label, and use the augmentation model to augment the graph $G$ to $G^-$ and form the negative graph pair $(G, G^-)$. The reward generation model is then trained to minimize the loss function $L = -\log s(G, G^+) - \log(1 - s(G, G^-))$, but the augmentation model is trained to maximize the reward $\log s(G, G^-)$ received from the reward generation model. In this adversarial training method, the reward generation model can actually be considered as the discriminator model. We conduct an exploration experiment of it on the COLORS and TRIANGLES datasets, but we find that this strategy cannot work well on both datasets. On the TRIANGLES dataset, gradient explosion consistently happens during the training but we have not yet figure out how to fix it. On the COLORS dataset, we show the learning curves of two models in Figure 6. Note that different from the learning curves of GraphAug in Figure 2, the augmentation model diverges and fails to learn to obtain more rewards as the training proceeds. In other words, the augmentation model struggles to learn to generate new graphs that can deceive the reward generation model. Given these existing problems in adversarial learning, we adopts the two-stage training pipeline in GraphAug and leaves the problem of simplifying the training to the future.

## F.3 COMPARISON WITH MANUALLY DESIGNED LABEL-INVARIANT AUGMENTATIONS

An interesting question is how does our GraphAug compare with the manually designed label-invariant augmentation methods (assuming we can design them from some domain knowledge)? We try answering this question by empirical studies on COLORS and TRIANGLES datasets. Since we explicitly know how the labels of graphs are obtained from their data generation codes, we can design some label-invariant augmentation strategies. We compare GraphAug with three designed label-invariant augmentation methods, which are based on MaskNF, DropNode, and PerturbEdge transformations intentionally avoiding damaging label-related information. Specifically, for the COLORS dataset, we compare with MaskNF that uniformly masks the node features other than the color feature, and DropNode that uniformly drops the nodes other than green nodes. In other words, they are exactly using the ground truth labels indicating which graph elements are label-related information, so we call them as MaskNF with GT and DropNode with GT. Note that no PerturbEdge with GT is defined on the COLORS dataset because the modification of edges naturally ensures label-invariance. Similarly, for the TRIANGLES dataset, we compare with DropNode with GT and PerturbEdge with GT which intentionally avoid damaging any nodes or edges in triangles.

The performance of no augmentation baseline, three manually designed augmentation methods, and our GraphAug method is summarized in Table 8. It is not surprising that all augmentation methods can outperform no augmentation baseline since they all can produce label-invariant training samples. Interestingly, GraphAug is a competitive method compared with these manually designed label-invariant methods. GraphAug outperforms manually designed augmentations on the COLORS dataset but fails to do it on the TRIANGLES dataset. We find that is because GraphAug model selects MaskNF with higher chances than DropNode and PerturbEdge, but graph classification models benefits more from diverse topology structures produced by DropNode and PerturbEdge transformations. **Note that although our GraphAug may not show significant advantages over manually designed label-invariant augmentations on these two synthetic datasets, in most scenarios, designing such label-invariant augmentations is impossible because we do not know which graph**

Table 9: The performance on seven benchmark datasets with the GCN model. We report the average ROC-AUC and standard deviation over ten runs for the ogbg-molhiv dataset, and the average accuracy and standard deviations over three 10-fold cross-validation runs for the other datasets.

| Method | PROTEINS | IMDB-BINARY | COLLAB | MUTAG | NCI109 | NCI1 | ogbg-molhiv |
|---|---|---|---|---|---|---|---|
| No augmentation | 0.711±0.003 | 0.734±0.010 | 0.797±0.002 | 0.803±0.016 | 0.742±0.004 | 0.731±0.002 | 0.761±0.010 |
| Uniform MaskNF | 0.716±0.001 | 0.723±0.006 | 0.802±0.002 | 0.765±0.017 | 0.734±0.005 | 0.729±0.004 | 0.745±0.011 |
| Uniform DropNode | 0.714±0.005 | 0.733±0.001 | 0.798±0.002 | 0.759±0.007 | 0.727±0.003 | 0.722±0.003 | 0.723±0.012 |
| Uniform PerturbEdge | 0.694±0.003 | 0.732±0.010 | 0.795±0.003 | 0.744±0.004 | 0.634±0.006 | 0.638±0.011 | 0.746±0.013 |
| Uniform Mixture | 0.714±0.003 | 0.734±0.009 | 0.797±0.004 | 0.754±0.004 | 0.731±0.002 | 0.722±0.002 | 0.743±0.011 |
| DropEdge | 0.710±0.006 | 0.735±0.013 | 0.797±0.004 | 0.762±0.003 | 0.724±0.004 | 0.723±0.003 | 0.757±0.012 |
| M-Mixup | 0.714±0.004 | 0.728±0.007 | 0.794±0.003 | 0.783±0.007 | 0.739±0.005 | 0.741±0.002 | 0.753±0.014 |
| $\mathcal{G}$-Mixup | 0.724±0.006 | 0.749±0.010 | 0.800±0.027 | 0.799±0.004 | 0.509±0.005 | 0.506±0.005 | 0.763±0.008 |
| FLAG | 0.723±0.003 | 0.743±0.008 | 0.797±0.002 | 0.819±0.004 | 0.746±0.003 | 0.734±0.004 | 0.768±0.010 |
| JOAOv2 | 0.722±0.003 | 0.687±0.010 | 0.681±0.004 | 0.736±0.007 | 0.691±0.007 | 0.672±0.004 | 0.722±0.009 |
| AD-GCL | 0.691±0.011 | 0.697±0.011 | 0.612±0.004 | 0.665±0.001 | 0.634±0.003 | 0.641±0.004 | 0.752±0.013 |
| AutoGCL | 0.668±0.008 | 0.719±0.002 | 0.745±0.002 | 0.769±0.022 | 0.707±0.002 | 0.714±0.005 | 0.701±0.014 |
| GraphAug | **0.736±0.007** | **0.764±0.008** | **0.808±0.001** | **0.832±0.005** | **0.760±0.003** | **0.748±0.002** | **0.774±0.010** |

Table 10: Average augmentation time per graph with the trained augmentation model.

| Method | PROTEINS | IMDB-BINARY | COLLAB | MUTAG | NCI109 | NCI1 | ogbg-molhiv |
|---|---|---|---|---|---|---|---|
| JOAOv2 | 0.0323s | 0.0854s | 0.2846s | 0.0397s | 0.0208s | 0.0223s | 0.0299s |
| AD-GCL | 0.0127s | 0.0418s | 0.1478s | 0.0169s | 0.0092s | 0.0083s | 0.0115s |
| AutoGCL | 0.0218s | 0.0643s | 0.2398s | 0.0256s | 0.0162s | 0.0168s | 0.0221s |
| GraphAug | 0.0073s | 0.0339s | 0.1097s | 0.0136s | 0.0075s | 0.0078s | 0.0106s |

**elements are label-related. However, our GraphAug can still work in these scenarios because it can automatically learn to produce label-invariant augmentations.**

### F.4 MORE EXPERIMENTAL RESULTS ON GRAPH BENCHMARK DATASETS

The performance of different augmentation methods on all seven datasets with the GCN model is presented in Table 9. Besides, to quantify and compare the computational cost of our method and some automated graph augmentation baseline methods on each dataset, we test the average time they use to augment each graph and summarize the average augmentation time results in Table 10. For most dataset, our method only takes a very small amount of time ($< 0.05$s) to augment a graph in average, which is an acceptable time cost for most real-world applications. In addition, from Table 10, we can clearly find that among all automated graph augmentation methods, our GraphAug takes the least average runtime to augment graphs. For the other baseline methods in Table 2, because they do not need the computation with neural networks in augmentations, their runtime is unsurprisingly lower ($< 0.001$s per graph). However, the classification performance of them is consistently worse than our GraphAug. Overall, our GraphAug achieves the best classification performance, and its time cost is the lowest among all automated graph augmentations.

### F.5 MORE ABLATION STUDIES

**Ablation on combining three different transformations.** In our method, we use a combination of three different graph transformations, including MaskNF, DropNode, and PerturbEdge. Our GraphAug model are designed to automatically select one of them at each augmentation step. Here we explore how the performance will change if only one category of graph transformation is used. Specifically, we compare with three variants of GraphAug that only uses learnable MaskNF, DropNode, and PerturbEdge, whose performance are listed in the first three rows of Table 11. We can find that sometimes using a certain category of learnable augmentation gives very good results, e.g., learnable DropNode on the NCI1 dataset. However, not all categories can achieve it, and actually the optimal category varies among datasets because graph structure distributions or modalities are very

Table 11: Results of ablation studies about combining three different transformations. We report the average accuracy and standard deviation over three 10-fold cross-validation runs with the GIN model.

| Method | PROTEINS | IMDB-BINARY | NCI1 |
|---|---|---|---|
| GraphAug with only learnable MaskNF transformation | 0.712±0.001 | 0.751±0.002 | 0.809±0.002 |
| GraphAug with only learnable DropNode transformation | 0.716±0.003 | 0.752±0.005 | 0.814±0.002 |
| GraphAug with only learnable PerturbEdge transformation | 0.702±0.009 | 0.754±0.005 | 0.780±0.001 |
| GraphAug | **0.722±0.004** | **0.762±0.004** | **0.816±0.001** |

Table 12: Results of ablation studies about using virtual nodes. We report the average accuracy and standard deviation over three 10-fold cross-validation runs with the GIN model.

| Method | PROTEINS | IMDB-BINARY | NCI1 |
|---|---|---|---|
| GraphAug with sum pooling | 0.711±0.005 | 0.750±0.008 | 0.788±0.004 |
| GraphAug with mean pooling | 0.711±0.004 | 0.752±0.004 | 0.801±0.005 |
| GraphAug with max pooling | 0.713±0.002 | 0.737±0.005 | 0.795±0.005 |
| GraphAug with virtual nodes | **0.722±0.004** | **0.762±0.004** | **0.816±0.001** |

Table 13: The label-invariance ratios on the test sets of COLORS and TRIANGLES datasets.

| Dataset | Uniform MaskNF | Uniform DropNode | Uniform PerturbEdge | Uniform Mixture | GraphAug |
|---|---|---|---|---|---|
| COLORS | 0.3547 | 0.3560 | 1.0000 | 0.5645 | 0.9994 |
| TRIANGLES | 1.0000 | 0.6674 | 0.1957 | 0.6181 | 1.0000 |

different in different datasets. Nonetheless, GraphAug can consistently achieve good performance without manually searching the optimal category on different datasets. Hence, combining different transformations makes it easier for the GraphAug model to adapt to different graph datasets than using only one category of transformation.

**Ablation on using virtual nodes.** In our method, virtual nodes are used to capture graph-level representation and predict augmentation categories due to two advantages. (1) First, in the message passing process of GNNs, virtual nodes can help propagate messages among far-away nodes in the graph. (2) Second, virtual nodes can learn to more effectively capture graph-level representations through aggregating more information from important nodes or structures in the graph (similar to the attention mechanism). In fact, many prior studies (Gilmer et al., 2017; Hu et al., 2020) have demonstrated the advantages of using virtual nodes in graph representation learning. To justify the advantages of using virtual nodes in GraphAug, we compare the performance of taking different ways to predict the augmentation category in an ablation experiment. Specifically, we evaluate the performance of GraphAug model variants in which virtual nodes are not used, but the augmentation category is predicted from the graph-level representations obtained by sum pooling, mean pooling, or max pooling. The results of them are summarized in the first three rows of Table 12. From the results, we can find that using virtual nodes achieves the best performance, hence it is the best option.

## F.6 EVALUATION OF LABEL-INVARIANCE PROPERTY

We evaluate the label-invariance ratios of our GraphAug method and the baseline methods used in Table 1 on the test sets of two synthetic datasets. The results are summarized in Table 13. Since the label is defined as the number of nodes with green colors (indicated by node features) in the COLORS dataset, Uniform DropNode and Uniform PerturbEdge will destroy label-related information and achieve a very low label-invariance ratio. Similarly, the label is defined as the number of 3-cycles in the TRIANGLES dataset, Uniform DropNode and Uniform PerturbEdge also achieve a

very low label-invariance ratio. However, our GraphAug can achieve a very high label-invariance ratio of close to 1.0 on both datasets. Besides, combining with the classification performance in Table 1, we can find that only the augmentations with high label-invariance ratios can outperform no augmentation baseline. This phenomenon demonstrates that label-invariance is significant to achieve effective graph augmentations.

