# OpenReview forum: "Automated Data Augmentations for Graph Classification"
_ICLR.cc/2023/Conference — ICLR 2023 poster_

### Official Review · Reviewer_jQkA · 2022-10-24

**Confidence:** 3
**Correctness:** 4
**Technical Novelty And Significance:** 4
**Empirical Novelty And Significance:** 3
**Recommendation:** 6

**Clarity, Quality, Novelty And Reproducibility:**

The paper is clear and well-written. It appears that the idea is novel with respect to prior work, and addresses an overlooked problem in GNN research. The extensive description of the method in the main manuscript and appendix should make it possible to easily reproduce the work.

Here are a few specific questions:
- The intuition of augmentation category transformations in section 3.4 is unclear. It seems like the p(a_t) is computed by aggregating information from all nodes and edges. Is this node/edge information computed by the reinforcement learning algorithm?
- Are there any steps taken to ensure stability of the reinforcement learning algorithm?
- It would help to see a discussion of the pros and cons of the effect of the fixed/pre-trained reward generation method.
- Why is it necessary to add a virtual node to the graph? Why is it not possible to predict the augmentation choice directly from the input graph?


**Strength And Weaknesses:**

The paper highlights and addresses an important problem in graph classification: how to perturb data without affecting label information. To address this problem, a reinforcement learning approach is proposed and extensive experiments demonstrate the utility of the technique. The paper is very well-written and easy to understand. Summary of related work is also extensive.

The key weakness is the lack of evaluation of the label invariance property. It’s not clear how often does the reinforcement learning method generate graphs that retain the same labels. Since this property is key to the motivation of the paper, I would like to see an analysis of how it is maintained.


**Summary Of The Paper:**

The paper proposed a reinforcement learning based approach for automatically predicting graph augmentations for a graph neural network (GNN) classification problem. The authors argue that label invariance (data augmentations that do not affect labels) is an important, unsolved problem for GNN, and propose a technique to maximize this property. Previous automatic graph augmentation methods did not consider labels, while the proposed method does. Experimental results demonstrate a marked improvement in classification accuracy on several benchmark datasets.


**Summary Of The Review:**

Overall, the paper should be a strong contribution to the conference. However, evaluation of the key component of the paper (label invariance) is missing. If the authors are able to address this concern, I will increase my rating.

EDIT: after reading through author rebuttals and other reviews, my concerns have been addressed and I increased my rating.

---

> ### Author Response · Authors · 2022-11-14
> **Responses to reviewer jQkA part 1/2**
>
> Thank you for your constructive feedback and comments. We believe your concerns and questions can be addressed by our responses below. We hope you can increase your score if we address your concerns.
>
> > **Q1: The key weakness is the lack of evaluation of the label invariance property. It’s not clear how often does the reinforcement learning method generate graphs that retain the same labels. Since this property is key to the motivation of the paper, I would like to see an analysis of how it is maintained.**
>
> Thank you for this insightful suggestion. We have conducted or added the following experiments and analyses about the label invariance property on the synthetic datasets.
>
> - First, as described in Section 4.1, to check whether our augmentation model can learn to produce label-invariant augmentations, we calculate the label-invariance ratio on the validation sets of two synthetic datasets at different training iterations. Note that **the label-invariance ratio is exactly defined as how often the method generates graphs that retain the same labels.** The changing curves of label-invariance ratios are visualized in Figure 2. These curves clearly show that on the COLORS and TRIANGLES dataset, as the training proceeds, our GraphAug model can gradually learn to produce augmentations leading to higher label-invariance ratios, i.e., generate graphs that retain the same labels with higher probability.
>
> - Second, we additionally evaluate the label-invariance ratios of our GraphAug method and the baseline methods used in Table 1 on the test sets of two synthetic datasets. The results are summarized in the following table. Since the label is defined as the number of nodes with green colors (indicated by node features) in the COLORS dataset, Uniform DropNode and Uniform PerturbEdge will destroy label-related information and achieve a very low label-invariance ratio. Similarly, the label is defined as the number of 3-cycles in the TRIANGLES dataset, Uniform DropNode and Uniform PerturbEdge also achieve a very low label-invariance ratio. However, our GraphAug can achieve a very high label-invariance ratio of close to 1.0 on both datasets. Besides, combining with the classification performance in Table 1, we can find that only the augmentations with high label-invariance ratios can outperform no augmentation baseline. This phenomenon demonstrates that label-invariance is significant to achieve effective graph augmentations. We have added these new results and analyses to Appendix F.6 in the revision.
>
> |Dataset|Uniform MaskNF|Uniform DropNode|Uniform PerturbEdge|Uniform Mixture|GraphAug|
> |----|----|----|----|----|----|
> |COLORS|0.3547|0.3560|1.0000|0.5645|0.9994|
> |TRIANGLES|1.0000|0.6674|0.1957|0.6181|1.0000|
>
> All the above experiments and analyses are conducted on synthetic datasets. We can calculate label-invariance ratios for them because we explicitly know which features or structures in graphs correspond to label-related information. However, this is not the case for graph benchmark datasets used in the experiments of Section 4.2, so we cannot do similar experiments or analyses for them. Nonetheless, we believe the consistent good performance of GraphAug over baseline methods in Table 2 demonstrates that GraphAug can also learn to produce label-invariant augmentations on graph benchmark datasets.
>
> > **Q2: The intuition of augmentation category transformations in section 3.4 is unclear. It seems like the $p(a_t)$ is computed by aggregating information from all nodes and edges. Is this node/edge information computed by the reinforcement learning algorithm?**
>
> No, the information is not computed by the reinforcement learning algorithm, but by neural networks (including the GNN encoder and MLPs described in Section 3.3) in the automated augmentation model. Specifically, neural networks first predict the probability distribution of augmentation categories and the exact augmentation category $c_t$ is sampled from this distribution. Based on what $c_t$ is, as described in the three bullets of Section 3.3, neural networks predict probability distributions of possible transformations for all graph elements in the graph (e.g., being dropped or kept for all graph nodes), then the exact transformation of each graph element is sampled from distributions. In Section 3.4, to calculate $p(a_t)$, the probability $p(c_t)$ that $c_t$ is sampled is multiplied with the product of the probabilities that the exact transformations of all graph elements are sampled. **Intuitively, $p(a_t)$ can be simply understood as the product of probabilities of all sampled random variables involved in the augmentation process.**

---

> > ### Public Comment · ~Zhuang_Jian_Zhi1 · 2023-03-19
> > **Questions about p (at) in Section 3.4**
> >
> > Hello, I did not understand the meaning of the calculated p (at) in Section 3.4 of your paper. Could you explain why this p (at) was calculated and the origin of the calculated formula? If you could reply to me, I would be extremely grateful.

---

> > > ### Author Response · Authors · 2023-04-04
> > > **Responses to Questions**
> > >
> > > Hi Zhuang Jian Zhi,
> > >
> > > Thank you for your interest in our work! The formula of $p(a_t)$ in Section 3.4 (Equation (4), (5), (6)) is calculating the probability that the augmentation transformation $a_t$ is sampled from the augmentation model. According to our description in Section 3.3, the augmentation transformation $a_t$ is sampled by first sampling the category of transformation $c_t$, then sampling the transformation performed on every graph element. Hence, $p(a_t)$ is calculated by multiplying $p(c_t)$ with the product of the transformation probabilities of all graph elements.
> > >
> > > Actually, the transformation performed on each graph element is sampled from Bernoulli distributions, and we can write its sampling probability in a closed form. Considering a Bernoulli distribution $B(p)$, and a sample $o\sim B(p)$ is sampled from this distribution. We know that the probability of $o=1$ is $p$ and the probability of $o=0$ is $1-p$. Then you can find that the probability of $o$, whatever $o$ is 0 or 1, can always be calculated as $p^o(1-p)^{1-o}$. By this trick, you can immediately understand that Equation (4), (5), (6) calculate the product of these closed-form probabilities over all graph elements (over all node features if $c_t$ is MaskNF, all nodes if $c_t$ is DropNode, and all edges if $c_t$ is PerturbEdge).
> > >
> > > Hope our explanations help you understand the details of calculating $p(a_t)$.
> > >
> > > Best,
> > > Authors

---

> ### Author Response · Authors · 2022-11-14
> **Responses to reviewer jQkA part 2/2**
>
> > **Q3: Are there any steps taken to ensure stability of the reinforcement learning algorithm?**
>
> We pre-train the reward generation model and make it fixed while training the augmentation model with the reinforcement learning algorithm. We find this is the most important step to ensure the training stability. No other steps are taken to stabilize the training.
>
> > **Q4: It would help to see a discussion of the pros and cons of the effect of the fixed/pre-trained reward generation method.**
>
> Thanks for this helpful suggestion.
>
> - The advantages of using the fixed/pre-trained reward generation model are two-fold. First, pre-training the reward generation model enables it to accurately predict whether two input graphs have the same labels or not, so that the generated reward signals can provide accurate feedback for the augmentation model. Second, using the fixed reward generation model can stabilize the training of the augmentation model in practice. As we shown in Appendix F.2, if the reward generation model is not fixed and jointly trained with the augmentation model, the training becomes very unstable.
>
> - The disadvantage of pre-training the reward generation model is that this training pipeline is time-consuming, because we have to train two models every time to obtain the finally usable graph augmenter.
>
> We have added the discussions of the pros and cons of the fixed/pre-trained reward generation model to the third paragraph of Appendix D in the revision.
>
> > **Q5: Why is it necessary to add a virtual node to the graph? Why is it not possible to predict the augmentation choice directly from the input graph?**
>
> It is not necessary to predict the augmentation category from the virtual node. Actually, the augmentation category can be predicted from any graph-level representation. We use the virtual node to capture graph-level representation due to two advantages. First, in the message passing process of GNNs, virtual nodes can help propagate messages among far-away nodes in the graph. Second, virtual nodes can learn to more effectively capture graph-level representations through aggregating more information from important nodes or structures in the graph (similar to the attention mechanism). In fact, many prior studies [ref1, ref2] have demonstrated the advantages of using virtual nodes in graph representation learning. To justify the advantages of using virtual nodes in GraphAug, we compare the performance of taking different ways to predict the augmentation category in an ablation experiment. Specifically, we evaluate the performance of GraphAug variants in which virtual nodes are not used, but the augmentation category is predicted from the graph-level representations obtained by sum pooling, mean pooling, or max pooling. All other settings are kept the same and the performance is evaluated by the classification accuracy on PROTEINS, IMDB-BINARY, and NCI1 datasets. The results are summarized in the following table. From the results, we can find that using virtual nodes achieves the best  performance, hence it is the best option. We have added the discussions and new ablation experiment results to Appendix F.5 in the revision.
>
> |Method|PROTEINS|IMDB-BINARY|NCI1|
> |----|----|----|----|
> |GraphAug with sum pooling|0.711$\pm$0.005|0.750$\pm$0.008|0.788$\pm$0.004|
> |GraphAug with mean pooling|0.711$\pm$0.004|0.752$\pm$0.004|0.801$\pm$0.005|
> |GraphAug with max pooling|0.713$\pm$0.002|0.737$\pm$0.005|0.795$\pm$0.005|
> |GraphAug with virtual nodes|**0.722$\pm$0.004**|**0.762$\pm$0.004**|**0.816$\pm$0.001**|
>
> [ref1]: Gilmer, J., Schoenholz, S. S., Riley, P. F., Vinyals, O., & Dahl, G. E. (2017, July). Neural message passing for quantum chemistry. In International conference on machine learning (pp. 1263-1272). PMLR.
> [ref2]: Hu, W., Fey, M., Zitnik, M., Dong, Y., Ren, H., Liu, B., ... & Leskovec, J. (2020). Open graph benchmark: Datasets for machine learning on graphs. Advances in neural information processing systems, 33, 22118-22133.

---

> > ### Comment · Reviewer_jQkA · 2022-11-15
> > **thank you**
> >
> > Thank you for the detailed explanations. I believe the evaluation of label invariance using synthetic data as well as the RL training insights are useful contributions thelselves.  I have updated my rating.

---

> > > ### Author Response · Authors · 2022-11-15
> > > **Thanks for your feedback.**
> > >
> > > Thank you very much for raising your rating! We are pleased to know that our responses have addressed your concerns.

---

### Official Review · Reviewer_2d1h · 2022-10-24

**Confidence:** 4
**Correctness:** 4
**Technical Novelty And Significance:** 3
**Empirical Novelty And Significance:** 3
**Recommendation:** 8

**Clarity, Quality, Novelty And Reproducibility:**

The key idea of the proposed methods is intuitive.
The proposed method is distinct from existing augmentation methods.
The paper presented the algorithms of proposed methods for reproducibility.


**Strength And Weaknesses:**

[Strength]
1. This paper studies a practical problem.
2. The main idea of the method is simple and intuitive.
3. The method achieves superior performance on various datasets for different classification tasks.
4. The overall presentation is good.
[Weakness]
1.   There are some missing prior works. For example, GCA [Zhu and Xu et al. WWW’21] proposed an adaptive graph augmentation for graph contrastive learning. Although it does not use “label-invariance” explicitly, it aims to keep important structures and attributes unchanged.
2.   The justifications for the reward generation network do not seem sufficient. The proposed method uses the reward generation network to generate the reward. However, in prior RL-based augmentation methods, the accuracy of the child network is given as the reward. So I think there should be justifications for the design decision of the network.


**Summary Of The Paper:**

This paper proposes GraphAug, a new automated graph augmentation method. The method selects augmentation operation and ratio, considering the label-invariance of the operation. The experimental results show that the proposed method improves accuracy on various graph classification tasks, compared to the baseline methods.


**Summary Of The Review:**

In summary, the paper studied a significant aspect of automated graph augmentation tasks, and the proposed method is based on intuitive and solid ideas. Although the idea of label-invariant augmentation exists in prior works, the design and the implementation of the model can be appreciated. Also, empirical results support its flexibility and superiority.

p.s.
Reference of GCA:
Yanqiao Zhu, Yichen Xu, Feng Yu, Qiang Liu, Shu Wu, and Liang Wang. 2021. Graph Contrastive Learning with Adaptive Augmentation. In Proceedings of the Web Conference 2021 (WWW '21), April 19–23, 2021, Ljubljana, Slovenia. ACM, New York, NY, USA 12 Pages.
https://dl.acm.org/doi/fullHtml/10.1145/3442381.3449802

---

> ### Author Response · Authors · 2022-11-14
> **Responses to reviewer 2d1h**
>
> Thank you very much for your positive rating and insightful comments! We hope your questions can be addressed by our responses below.
>
> > **Q1: There are some missing prior works. For example, GCA [ref1] proposed an adaptive graph augmentation for graph contrastive learning. Although it does not use “label-invariance” explicitly, it aims to keep important structures and attributes unchanged.**
>
> Thank you very much for making us aware of this important prior work. The motivation of GCA [ref1] is that preserving important structures or node features is significant in designing effective graph augmentations, which is actually very similar to the motivation of our GraphAug. GCA identifies import edges and node features in the graph by node centralities, and augments the graph by random edge dropping and node feature masking, but assigns lower perturbation probabilities to the identified important edges and node features. In other words, **GCA is based on some manually designed rules to augment graphs.** However, our GraphAug method does not aim to define a fixed graph augmentation strategy for every graph. Instead, it seeks to **make the augmentation model find good augmentation strategies automatically** with reinforcement learning based training. We have added these discussions about GCA to the sixth paragraph of Appendix D in the revision.
>
> [ref1]: Zhu, Y., Xu, Y., Yu, F., Liu, Q., Wu, S., & Wang, L. (2021, April). Graph contrastive learning with adaptive augmentation. In Proceedings of the Web Conference 2021 (pp. 2069-2080).
>
> > **Q2: The justifications for the reward generation network do not seem sufficient. The proposed method uses the reward generation network to generate the reward. However, in prior RL-based augmentation methods, the accuracy of the child network is given as the reward. There should be justifications for the design decision of the network.**
>
> Thank you very much for this insightful suggestion. In our opinion, using the label-invariance probability predicted by the reward generation model as the reward has the following advantages over using the classification accuracy.
>
> - First, maximizing the label-invariance probability can directly encourage the augmentation model to produce label-invariant augmentations. However, the classification accuracy is not directly related to label-invariance, so using it as the reward feedback does not make the augmentation model learn to ensure label-invariance.
>
> - Second, predicting the label-invariance probability only needs one simple model inference process that is computationally cheap, while obtaining the classification accuracy is computationally expensive because it needs to train a classification model from scratch.
>
> - Most importantly, our reward generation scheme facilitates the learning of the augmentation model by **providing the reward feedback for every individual graph.** Even in the same dataset, the label-related structures or patterns in different graphs may vary a lot, hence, good augmentation strategies for different graphs can be different. However, the classification accuracy evaluates the classification performance when using the produced graph augmentations to train models on the overall dataset, which does not provide any feedback about whether the produced augmentation on every individual graph sample is good or not. Differently, the label-invariance probability is computed for every individual graph sample, thereby enabling the model to capture good augmentation strategies for every individual graph.
>
> We think the above three advantages provide sufficient justifications for taking the label-invariance probability predicted by the reward generation model rather than the classification accuracy as the reward. We have added these discussions to the fifth paragraph of Appendix D in the revision.

---

> > ### Comment · Reviewer_2d1h · 2022-11-25
> > **thank you**
> >
> > Thanks for the detailed response! I'll also keep my recommendation.

---

> > > ### Author Response · Authors · 2022-11-25
> > > **Thanks for your responses.**
> > >
> > > Thank you very much for your responses! We are pleased to know that you will keep your positive recommendation.

---

### Official Review · Reviewer_iXAY · 2022-10-24

**Confidence:** 4
**Correctness:** 3
**Technical Novelty And Significance:** 3
**Empirical Novelty And Significance:** Not applicable
**Recommendation:** 8

**Clarity, Quality, Novelty And Reproducibility:**

Clarity: This paper is overall well-written and easy to follow.

Quality: The proposed method seems sound to me.

Novelty: While the idea of using RL for auto-augmentation is not novel, the newly proposed label-invariant augmentation is novel and interesting.

Reproducibility: Pseudo-codes are provided in the appendix, but the actual code seems not provided.

**Strength And Weaknesses:**

S1. This paper is overall well-written and easy to follow.

S2. Extensive experiments on multiple datasets with SOTA baselines validates the effectiveness of the proposed method.

S3. The proposed label-invariant augmentations seems very interesting.

W1. I appreciate the complexity analysis in the appendix. It would be nice to also have runtime comparison with baselines to further validate the efficiency of the proposed method.

W2. In Table 2, sometimes the mixture augmentation would underperform some of the single augmentations. I wonder if the authors have conducted sufficient hyperparameter search for it.

W3. On the larger datasets such as the OGB one, the performance improvements over simple augmentations are not very big. In that case, I wonder if adding the much more complex RL framework would worth the trade-off on time and resource.

**Summary Of The Paper:**

This paper worked on data augmentation for graph classification, and proposed an automated method that learns label-invariant augmentations.

It formulates the graph augmentation problem as a sequential decision-making task (Markov Decision Process), where at each time step one augmentation method is picked given the current graph and transformed graph (as well as the reward). The model is optimized using policy gradient (i.e., REINFORCE) algorithm and the reward is generated using the graph matching network to predict the probability that the current graph and the transformed graph have the same label (so as to ensure label-invariant property). Experiments on both synthetic datasets and real-world benchmarks verify the effectiveness of the proposed method.

**Summary Of The Review:**

Based on my previous comments, I recommend acceptance.

---

> ### Author Response · Authors · 2022-11-14
> **Responses to reviewer iXAY**
>
> Thank you for your positive feedback and valuable comments! We hope your concerns and questions can be addressed by the following responses.
>
> > **Q1: It would be nice to also have runtime comparison with baselines to further validate the efficiency of the proposed method.**
>
> Thank you very much for this helpful suggestion. For the three automated graph augmentation methods (JOAOv2, AD-GCL, AutoGCL) in Table 2, we test the average time they use to augment each graph. The runtime comparison between our method and these methods are summarized in the following table.
>
> |Method|PROTEINS|IMDB-BINARY|COLLAB|MUTAG|NCI109|NCI1|ogbg-molhiv|
> |----|----|----|----|----|----|----|----|
> |JOAOv2|0.0323s|0.0854s|0.2846s|0.0397s|0.0208s|0.0223s|0.0299s|
> |AD-GCL|0.0127s|0.0418s|0.1478s|0.0169s|0.0092s|0.0083s|0.0115s|
> |AutoGCL|0.0218s|0.0643s|0.2398s|0.0256s|0.0162s|0.0168s|0.0221s|
> |GraphAug|**0.0073s**|**0.0339s**|**0.1097s**|**0.0136s**|**0.0075s**|**0.0078s**|**0.0106s**|
>
> From the above results, we can clearly find that among all automated graph augmentation methods, our GraphAug takes the least average runtime to augment graphs. For the other baseline methods in Table 2, because they do not need the computation with neural networks in augmentations, their runtime is unsurprisingly lower ($<0.001$s per graph in average). However, the classification performance of them is consistently worse than our GraphAug. Overall, our GraphAug achieves the best classification performance, and its time cost is the lowest among all automated graph augmentations. We have added the runtime results and discussions to Appendix F.4 and Table 10 in the revision.
>
> > **Q2: In Table 2, sometimes the mixture augmentation would underperform some of the single augmentations. I wonder if the authors have conducted sufficient hyperparameter search for it.**
>
> Yes, we have conducted sufficient hyperparameter search for the mixture augmentation baseline. As for the phenomenon that the mixture augmentation sometimes underperforms some single augmentations, we think it is due to the large negative impact from some bad augmentations of all the mixed augmentations. For instance, on the NCI1 dataset, using Uniform PerturbEdge to augment graph consistently produces bad training data samples so it leads the classification model to achieve the lowest classification accuracy. When using Uniform Mixture as the augmentation, one of Uniform MaskNF, Uniform DropNode, and Uniform PerturbEdge will be randomly selected to augment graphs each time. Hence, the classification model will be negatively affected by the bad samples produced from Uniform PerturbEdge, and even achieve worse performance than only using Uniform MaskNF or Uniform DropNode when the negative impact is large enough. Nonetheless, although the mixture augmentation cannot outperform all single augmentations in Table 2, it can outperform the single augmentation with the worst performance.
>
> > **Q3: On the larger datasets such as the OGB one, the performance improvements over simple augmentations are not very big. In that case, I wonder if adding the much more complex RL framework would worth the trade-off on time and resource.**
>
> We agree that applying GraphAug to large datasets is not worth the large time and other resource cost. On the ogbg-molhiv dataset, we find it takes the total time of around 10 hours to train the reward generation model and augmentation model before we obtain a finally usable graph augmenter. Given that the performance improvement is not significant on the ogbg-molhiv dataset, such a large time cost is not a worthwhile investment. **Our GraphAug mainly targets on improving the graph classification performance by generating more training data samples for the tasks with small datasets, particularly for those that need huge cost to manually collect and label new data samples.** But for the classification task with sufficient training data, the benefits of using GraphAug are limited and not worth the large time and resource cost to train GraphAug models. We have added these discussions to the fourth paragraph of Appendix D in the revision.
>
> > **Q4: Pseudo-codes are provided in the appendix, but the actual code seems not provided.**
>
> We do not provide the source codes of our method now because our institution has not yet approved us to make the source codes public. We will definitely open source our codes on GitHub after we receive the approval.

---

> > ### Comment · Reviewer_iXAY · 2022-11-23
> > **thank you**
> >
> > Thanks for the detailed response! I'll keep my recommendation.

---

> > > ### Author Response · Authors · 2022-11-24
> > > **Thanks for your feedback.**
> > >
> > > Thank you very much for your feedback! We are glad to know that you will keep your positive rating.

---

### Decision · Program_Chairs · 2023-01-20

**Decision:**

Accept: poster

**Justification For Why Not Higher Score:**

My main concern is that the proposed method may not be computationally efficient enough for data augmentation in practice.


**Justification For Why Not Lower Score:**

The paper is both novel and significant with strong experiment results. It is hard to reject such a paper.


**Metareview: Summary, Strengths And Weaknesses:**

While most existing automated data augmentation methods are for image classification, this paper is novel in that it proposes a method for graph classification which, as of now, has not benefited from automated data augmentation. Although the main idea of the proposed method is simple, it gives good results as demonstrated by extensive experiments. The paper is well written and easy to follow. The authors are recommended to consider the efficiency and hence practicality of the proposed method as it is one important factor that determines whether it will indeed be used for data augmentation in practice. The authors are also highly recommended to consider our comments and suggestions to revise their paper before publication.


**Note From Pc:**

if the above contains the word "oral" or "spotlight" please see: "oral" presentation means -> notable-top-5% and "spotlight" means -> notable-top-25%. As stated in our emails, we are disassociating presentation type from AC recommendations